# The underappreciated potential of peatlands in global climate change mitigation strategies

J. Leifeld [1] & L. Menichetti[1,2]

Soil carbon sequestration and avoidable emissions through peatland restoration are both strategies to tackle climate change. Here we compare their potential and environmental costs regarding nitrogen and land demand. In the event that no further areas are exploited, drained peatlands will cumulatively release 80.8 Gt carbon and 2.3 Gt nitrogen. This corresponds to a contemporary annual greenhouse gas emission of 1.91 (0.31–3.38) Gt $CO_2$-eq. that could be saved with peatland restoration. Soil carbon sequestration on all agricultural land has comparable mitigation potential. However, additional nitrogen is needed to build up a similar carbon pool in organic matter of mineral soils, equivalent to 30–80% of the global fertilizer nitrogen application annually. Restoring peatlands is 3.4 times less nitrogen costly and involves a much smaller land area demand than mineral soil carbon sequestration, calling for a stronger consideration of peatland rehabilitation as a mitigation measure.

---

[1] Agroscope, Climate and Agriculture Group, Reckenholzstrasse 191, 8046 Zurich, Switzerland. [2] Now Ecology, SLU (Sveriges Lantbruksuniversitet), Ulls Väg 16, 75651 Uppsala, Sweden. Correspondence and requests for materials should be addressed to L.M. (email: jens.leifeld@agroscope.admin.ch)

In intact peatland ecosystems, oxygen deficiency resulting from high-water tables causes the formation of organic soils[1]. These are distinguished from mineral soils by their high carbon (C) and nitrogen (N) density, often with an organic matter content of >90% and thicknesses of up to several meters[2,3]. Peatlands only account for ~3% of the terrestrial surface[4], predominately occurring in boreal and temperate ecosystems, with a smaller proportion in tropical regions. Nevertheless, they may store ~644 Gt of C[4–6] or 21% of the global total soil organic C stock of ~3000 Gt (0–3 m, ref. [7]). In addition, peatlands are large stores of organic N: Northern peatlands, characterized by wide C/N ratios between 12 and 217[8], have accumulated 8–15 Gt N[9], whereas the N stock in tropical peatlands has not yet been reviewed.

At present, human activity is either draining or mining ~10% of global peatlands[10], transforming them from long-term C sinks into sources by acting on three C loss pathways: $CO_2$ from microbial peat oxidation, dissolved C leaching, and $CO_2$, CO and $CH_4$ from peat fires and combustion of mined peat[11]. In addition, drained peatlands release relevant amounts of $N_2O$[11] and therefore drainage induces peatland degradation and alters peatlands, globally, from a net sink to a net source of greenhouse gas (GHG) in the land-use sector[12,13]. Consequently, peatland protection and restoration are seen as proximate mitigation measures[10,14]. Restoration through rewetting can significantly reduce GHG emissions[15], restore vegetation communities, and recover biodiversity[16], while still allowing for extensive management such as paludiculture[17–19].

A substantial fraction of anthropogenic GHG emissions could be compensated for by improving management of mineral soils. Indeed, various options for C sequestration, such as residue management and improved crop rotations in arable land or species introduction in grasslands, have been discussed (e.g., refs. [20,21]). However, stoichiometric constraints limiting C storage are more severe in mineral soils compared to peatlands. In organic soils, mostly phenolic or lignocellulosic remains from mosses or vascular plants with low concentrations of N and other nutrients predominate. In mineral soils, microbial products with narrow C/N ratios are quantitatively dominant components of organic matter[22], which makes C storage in mineral soil nutrient costly. C storage in peatlands is much less N limited: they accumulate organic matter over millennia, needing an average of only 0.018 kg N per kg C sequestered[8,9], which is considerably less than that of mineral soils at 0.094 kg N per kg C[23]. Hence, the well-recognized long-term climate cooling effect of global peatlands[24] comes at a relatively low N cost.

Here, we begin by revisiting data on global peatland distribution and degradation, then analyze current and future GHG release and its uncertainty from degrading peatlands (neglecting peat fires). This provides a global estimate of the magnitude of possible GHG savings for organic soils, assuming that peatland restoration renders a GHG neutral ecosystem. We compare this potential to management-induced organic matter accumulation in mineral soil and assess both pathways from the perspectives of C and N cycling, time, and land demand. Our analysis assigns higher GHG emissions to degraded peatlands than previous reports, mostly owing to a larger area of managed organic soils in the tropics. Cumulative GHG emissions from already drained peatlands are greater than mineral soil carbon sequestration potentials on all agricultural land. Restoration of degraded peatlands provides an efficient mitigation measure in the land-use sector, owing to a smaller area and nitrogen demand when compared with mineral soils.

## Results

**Global peatland area and GHG emissions.** Our maps of the peatland distribution (Figs. 1 and 2) reveal a contribution of 83.3, 4.0, and 12.7% from the boreal (and polar), temperate, and tropical zone, respectively, to the total peatland area. Our area estimate of in total 463.2 Mha puts a higher weight on tropical peatlands compared with previous estimates (8–11%, refs. [4,5]), owing to an updated climate classification (Methods) and inclusion of recent data[6]. It should be noted that the area estimate of global peatland is in general highly uncertain. The assignment of actual emission factors for drained organic soils[11], classified by climate and land use, to the overall peatland area is considered a vulnerability indicator and illustrates that potential hot spots for GHG emissions are mostly located in the tropics (Fig. 1). The global peatland area estimated as drained for forestry, cropland or grassland is ca. 50.9 Mha (Table 1). The distribution of actual and potential emissions differs substantially, particularly in the tropical zone of South America and Africa, indicating a low degree of degradation, whereas actual rates are closer to the potential

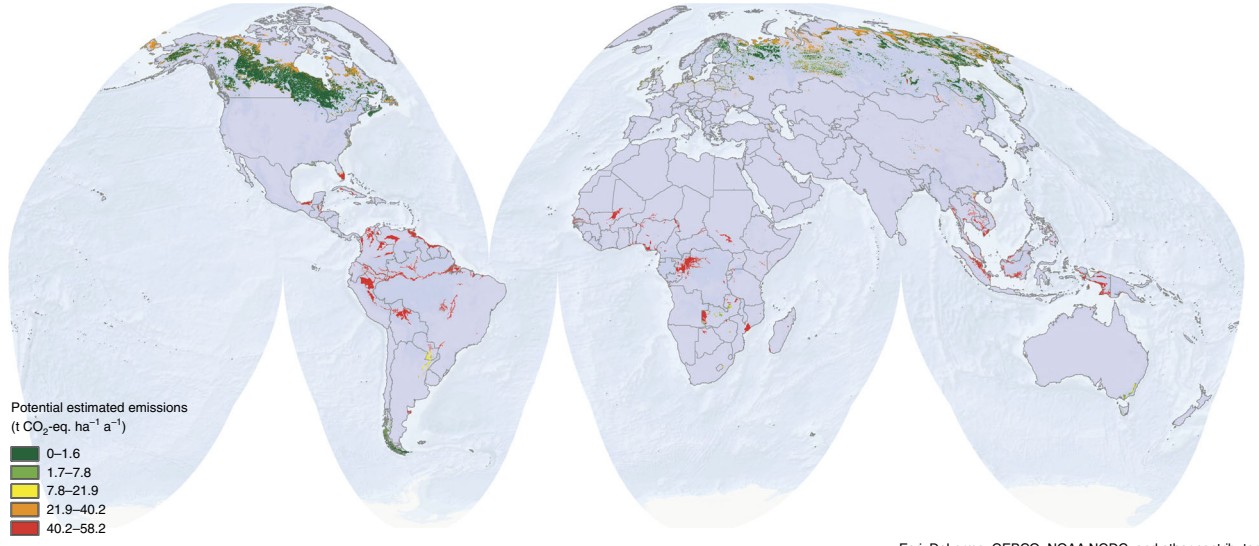

Potential estimated emissions
(t $CO_2$-eq. ha$^{-1}$ a$^{-1}$)

- 0–1.6
- 1.7–7.8
- 7.8–21.9
- 21.9–40.2
- 40.2–58.2

Esri, DeLorme, GEBCO, NOAA NGDC, and other contributors

**Fig. 1** Global peatland distribution and annual potential emissions from peatland degradation. The colored area indicates the areal distribution of peatlands globally. Legend colors refer to per hectare greenhouse gas emissions if those peatland would be drained. High potential emissions are associated with intensive land use and warm climate, low potential emissions with low land-use intensity in cold climate

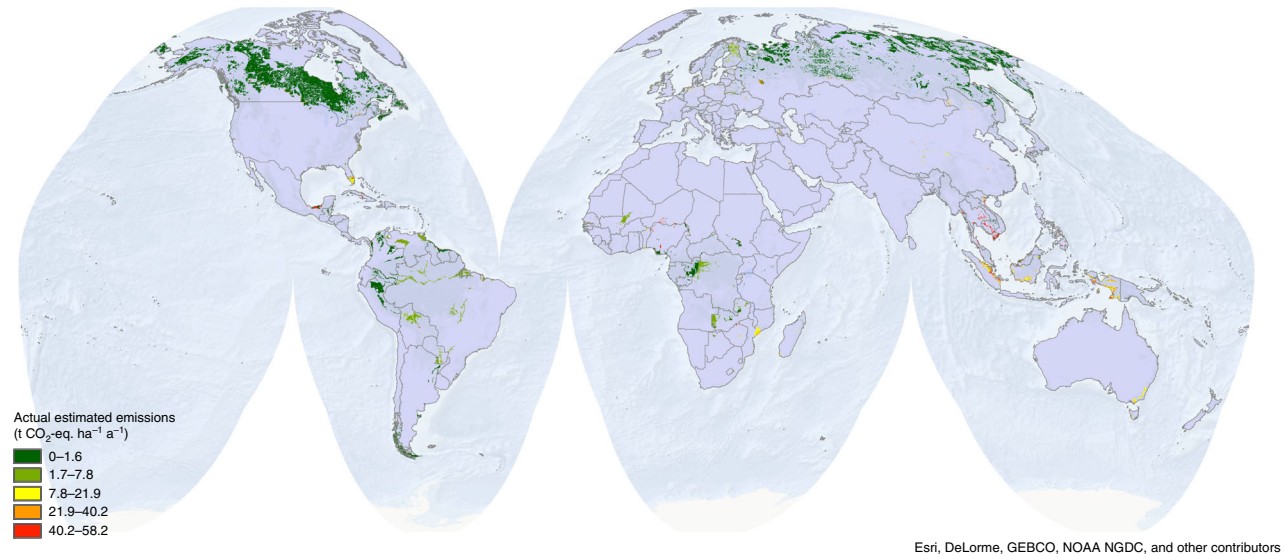

Actual estimated emissions
(t CO$_2$-eq. ha$^{-1}$ a$^{-1}$)

- 0–1.6
- 1.7–7.8
- 7.8–21.9
- 21.9–40.2
- 40.2–58.2

Esri, DeLorme, GEBCO, NOAA NGDC, and other contributors

**Fig. 2** Global peatland distribution and estimated annual actual emissions from peatland degradation. Data are calculated by correcting the potential emissions in Fig. 1 for the share of degraded peatland (see Supplementary Fig. 1), thereby highlighting regions with a large degree of disturbance. The map shows the average emission per area peatland for any given land use and climate category, not the average emission per area of disturbed peatland

### Table 1 Area and emission overview of global peatlands

| Climate | Total peatland | CL[a] | GL[a] | FL[a] | GL/FL | CL/GL/FL | Degrading peatland | Actual emissions[b] | Peat C | Degrading peat C |
|---|---|---|---|---|---|---|---|---|---|---|
| | Area (Mha) | | | | | | | Gt CO$_2$ eq. | Gt C | |
| Tropical | 58.7 | 8.5 | 11.3 | 34.6 | 1.6 | 2.7 | 24.2 | 1.48 (0.04–2.79) | 119.2[c, e] | 49.1 |
| Temperate | 18.5 | 3.5 | 5.0 | 8.9 | 0.7 | 0.6 | 10.6 | 0.16 (0.10–0.21) | 21.9[d, e] | 12.5 |
| Boreal | 360.9 | 6.8 | 85.6 | 249.5 | 17.9 | 1.1 | 15.5 | 0.26 (0.16–0.36) | 427.0[d, e] | 18.3 |
| Polar | 25.0 | 0.1 | 14.9 | 9.7 | 0.1 | 0.3 | 0.7 | 0.01 (0–0.02) | 29.6[d, e] | 0.8 |
| Oceanic | <0.1 | <0.1 | <0.1 | <0.1 | <0.1 | <0.1 | <0.1 | 0 (0–0) | <0.1 | <0.1 |
| Total | 463.2 | | | | | | 50.9 | 1.91 (0.31–3.38) | 597.8 | 80.8 |

[a]Peatland area distributed by land use. Land-use classes are cropland (CL), grassland (GL) and forest land (FL). Rows with more than one land-use represent areas where a clear assignment to one type is not possible (see Methods)
[b]Annual means, values in parentheses express the range from minimum to maximum. Emissions include CO$_2$, CH$_4$, N$_2$O, and DOC
[c]Based on refs. [5,6]
[d]Based on ref. [8]
[e]Assignment to climate region by applying C densities in refs. [5,6,8] to our newly delineated areas

ones in Southeast Asia and Europe, owing to their higher degree of degradation[25,26]. Degraded peatlands store globally ~80.8 Gt soil C and emit ~1.91 (0.31–3.38) Gt CO$_2$-eq. a$^{-1}$ (0.52 (0.08–0.92) Gt CO$_2$-C-eq. a$^{-1}$) mostly as CO$_2$ (Table 1 and Methods). Our analysis categorizes degraded peatlands by climate and shows that, in accordance with previous studies[27,28], the largest GHG emitters are drained tropical peatlands (Table 1, Supplementary Fig. 1).

Depending on the quantity of C stocks, GHG emission factors, and contribution of C to the overall emission, a drained peatland continues to emit for decades to centuries. The ongoing organic matter loss will exhaust the actual degrading peat deposits within the next centuries (Fig. 3), depending on the intensity of the emissions and their C density. We calculated a N stock in tropical peat (intact and degrading) of ~4.0 Gt N, based on a median C/N ratio of 29.7 (Methods) and a tropical peat deposit of 119.2 Gt C. This latter estimate is based on our revised, high-resolution area assignment and comprehensive estimates on carbon stocks per area land in tropical peat[4,5,6]. A discovery as in ref. [6] suggests that the tropics might present still uncharted peat stocks, for example in the Amazon basin. Of this N in tropical peatlands, 1.7 Gt

would be cumulatively released upon degradation of currently utilized tropical peatlands, together with 0.6 Gt N from currently degrading temperate, boreal, and polar peatlands (total N stock temperate + boreal + polar peatlands 9.8 Gt, median C/N 49.0[8]). In reverse, these quantities of C and N stocks in degrading peatlands represent avoidable emissions during the calculated time span and become an asset upon restoration.

Even the most recent global estimate on the contemporary net biogenic terrestrial CO$_2$ sink of −5.3 (±4.5) Gt CO$_2$ a$^{-1}$ does not fully consider peatland emissions[29]. Hence, our analysis suggests the magnitude of that sink is probably undervalued. We use, in contrast to any of the previous estimates, the most recent Intergovernmental Panel on Climate Change (IPCC) emission factors[11], a map of peatland area updated to 2016 and including areas not previously considered, a refined allocation of drained organic soils to land use and climate regions and a revised estimate of degrading peatland areas based on such allocation (Methods). For the first time with regard to such estimates, our calculation accounts for all GHGs (CO$_2$, N$_2$O, CH$_4$, CO$_2$ from oxidizing leached dissolved organic C (DOC) (Methods). The upper end of the range of our estimate for annual GHG emissions

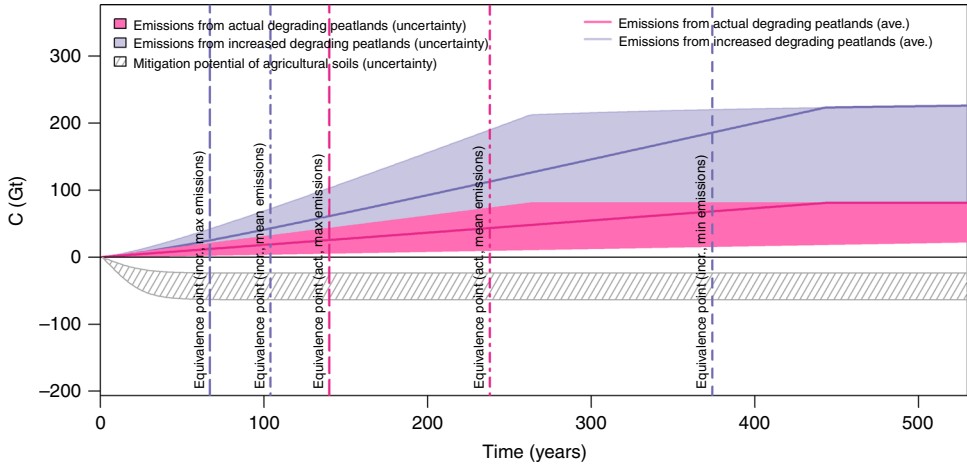

**Fig. 3** The world's cumulative GHG emissions from degrading peatlands. Cumulative emissions of the current area are in pink, and for a doubling of that area within 60 years (future scenario, see Methods) in blue (positive values). Their end points represent the theoretical maximum that can be emitted from the degrading peat C stocks. Units are Gt $CO_2$-C-eq. Negative areas display the cumulated C sink in mineral soils assuming the C saturation equation proposed by ref. [31] (gray). Vertical lines denote equivalence points where mineral soil sink potentials are exhausted relative to peatland-related emissions (minimum, average and maximum estimates, for numbers see text; maximum equivalence point for actual degrading peatland is 1021 years and not displayed)

from drained organic soils is higher than previous studies reporting 0.35 Gt $CO_2$-C-eq. $a^{-1}$ (ref. [10], all drained peatlands, $CO_2$ only, area 46 Mha), 0.24 Gt $CO_2$-C-eq. $a^{-1}$ (ref. [30], drained for cropland and grassland, $CO_2$ from cropland, $N_2O$ from cropland and grassland, area 26 Mha), or 0.25 Gt $CO_2$-C-eq. $a^{-1}$ (ref. [28], drained for agriculture, $CO_2$ and $N_2O$, area 25 Mha) per year. This difference arises despite smaller EF's in the most current report[11], as compared to ref. [10], mainly because of a higher fraction allocated to tropical peatlands known for their much higher emission factors, and including all GHGs and land-use types.

**Mitigation potentials in mineral and organic soils**. Annual mitigation potentials for cropped organic soil restoration, considering all GHGs, were previously estimated at 0.08–0.35 Gt $CO_2$-C-eq. $a^{-1}$, mostly stemming from avoided $CO_2$ emissions[20,21]. This potential is here adjusted upward (0.08–0.92 Gt $CO_2$-C-eq. per year), although estimates mostly agree regarding the lower boundary. We consider the upper side of this range as maximum mitigation potential achievable with full global peatland restoration, and we account for the full uncertainty range in the following calculations.

How does the global GHG saving potential of peatland restoration compare with that of mineral soil abatement strategies? Current estimates indicate sequestration potentials in agricultural soils of up to 64 Gt C over the next century[31] i.e., of the same order of magnitude as avoidable emissions from the currently degrading peatland C pool. The 64 Gt potential considers mitigation measures on all 4923 Mha agricultural land[32] and it is the most optimistic scenario. We adopted the principle of ref. [31] for calculating sequestration potentials of agricultural mineral soils and, within our simulation, the maximum additional C storage of 24–64 Gt C in 4923 Mha mineral soil would be reached after 63 years (Methods). We did not consider any increase of the current agricultural area, since any land-use change would imply net GHG release from newly converted areas. Our calculation suggests that, in a hypothetic scenario in which substantial peatland restoration is not incorporated, comprehensive mineral soil measures alone would only be able to compensate for most of the future emissions from degrading organic soils but not provide a net soil sink. A detailed

look into the time line of emissions reveals that, because of the high-annual GHG release and the size of the peat C pool, the equivalence point after which cumulative mineral soil C sequestration becomes counterbalanced by cumulative GHG emissions from degrading peatlands would be reached after on average 238 (140–1021) years (Fig. 3). This will apply even if sequestration strategies were implemented at full capacity in mineral soils and considering that no additional peatland areas would be degraded. After this point, GHG emissions from soils would globally represent a net GHG source to the atmosphere. In a scenario of doubling peatland exploitation area gradually during the next six decades (Methods), the equivalent point would be reached after only 104 (67–374) years, and cumulative C and N release from that larger area would add up to 161.6 and 4.6 Gt, respectively.

**Assessment of soil related mitigation measures**. Minimizing anthropogenic GHG emissions via peatland restoration is cheap in terms of N when compared with mineral soil C sequestration and thus more cost effective. Owing to their narrow C/N ratio of on average 10.7[23], N availability will represent a constraint to compensate for anthropogenic GHG emissions in mineral soil together with making such measurements more pricey. This can be illustrated by comparing the avoidable N release from currently degrading peatlands with the N requirement for mineral soil C sequestration; The latter would entail a total of 2.2–5.9 Gt N or, averaged over 63 years, 35.2–93.9 Mt N $a^{-1}$, to be immobilized in agricultural mineral soils to offset GHG emissions of the same magnitude from other sources as new organic matter storage. For perspective, with a quantity of 2.2–5.9 Gt N, global peatlands sequestered 81–216 Gt C. The additional N requirement for organic matter sequestration in mineral soils is substantial considering that 117 Mt N were applied globally as fertilizer in 2016[33], and stands in sharp contrast to halting emissions via organic soil restauration where no additional N must be fixed but quite the converse, release of 2.3 Gt N can be avoided. Taking the mean equivalence point of 238 years in the business as usual scenario of Fig. 3, this number converts to annually 9.7 Mt N, whose release can be circumvented by peatland restoration.

Of course, mineral soil C sequestration is not a purpose in itself, but a co-benefit of practices applied to improve soil health and, hence, productivity[34]. Still, and although not all C sequestration measures in mineral soil require N fertilization explicitly, the unavoidable microbial transformation of wide-C/N-ratio plant litter into narrow-C/N-ratio SOM implies co-sequestration of C and N[35]. It is also reasonable to assume that this additional N input comes with costs in form of energy for N fertilizer production[36] and N losses to the environment[37], including additional $N_2O$[38] (which will need other mitigation interventions). In addition, a vast area of agricultural land on mineral soils needs improved management with continuous maintenance owing to the non-permanence of mineral soil C storage[21,39]. By integrating temporal dynamics instead of just comparing annual rates, we show that even in the case of successful full implementation of C sequestration measures in all 4923 Mha agricultural land on mineral soil, GHG emissions from only the 50.9 Mha of currently degrading peatlands would override that sink on a midterm time scale. Hence, the global perspective on soil-born GHG emissions reveals that a net mitigation cannot be achieved without including the organic soil option. In a scenario of further peatland exploitation, a positive global soil GHG balance, i.e., a net source, would likely be reached within the next one hundred years.

Our comparative simulation of mineral soil C sequestration and GHG mitigation from organic soil underpins the high relative mitigation efficiency of peatland restoration and protection. The majority of utilized peatlands is located in the tropical (47% of the global total degrading area) and boreal+polar (32% of the global total degrading area) zones, followed by a relevant percentage of utilized peatland in temperate zones (21%), and many of these areas are managed at high intensity. A full recovery of these areas implies that the corresponding loss of production from agriculture and forestry should be compensated by an increased productivity on mineral soil. Paludiculture allows for maintaining a non-intensive land use, but cannot fully replace food production. However, the 50.9 Mha of degrading peatlands represents only ~1% of total agricultural land. Under the assumption that productivity on degrading organic soils is similar to that of mineral soil[40] and that all of them were agriculturally used, it is unlikely that the environmental impact of increasing productivity per land unit on mineral soil by 1% overrides the detrimental consequences of continuous peatland use listed here.

These insights into mitigation options in organic and mineral soils in terms of N and land demand, and time line do not call into question efforts undertaken in mineral soils, but indicate that, for effective climate change mitigation in the land-use sector at the global scale, both strategies must be adopted and that we must prevent further peatland degradation. Integrating significant peatland preservation and restoration measures in global policies, particularly in the tropics, therefore appears to be a critical step for an effective global climate change mitigation strategy.

## Methods

**Data sources for the peatland extension map**. The map we built represents an upper estimate of the possible global peatland area and its spatial distribution, in order to offer a precautionary estimate. There are still many uncharted peatlands are in the world, particularly in the tropical area[6], and we therefore assembled the global map of potential peatland areas starting from the map published in ref. [4] (please note that such map relies on different sources than the calculations reported in the same paper and considers a substantially larger area), and updated it with the most recent data from the literature. We added all the areas considered in the Ramsar convention (http://www.ramsar.org/sites-countries/the-ramsar-sites) and included country-specific data: The data from Sweden have been received from the Swedish Geological Union. The data for Estonia were received from the Estonian Land Board, Department of Geology. The data for Kirghizistan have been taken from ref. [41]. The important areas of Malaysia, Sumatra, and Borneo were updated with the maps from ref. [42]. The data for Tasmania were received directly from the

Tasmanian Department of Primary Industries, Parks, Water, and Environment[43]. Our updated peatland extension map accounted for an increase of 0.8% compared to ref. [4]. We utilized our map only as a statistically representative geographical sample to recalculate proportions of emissions and land use based on other spatial data.

**The peatland land use and degradation maps**. All the work with spatial data has been performed with ArcGIS (ArcGIS 10.4, ESRI, Redlands, CA, USA). All calculations adopt a Goode homolosine projection in order to have approximately equal area per pixel.

We assigned peatland areas to the three major IPCC land-use classes (Forest Land FL, Grassland GL, and Cropland CL) by overlaying our map with the European Space Agency Global Land Cover Map (ESA-GLCM, release 19/12/2008), available at http://www.esa.int/spaceinimages/Images/2008/12/Envisat_global_land_cover_map (accessed 21 Jan 2016). The ESA-GLCM raster distinguishes 23 classes (Supplementary Table 1) that we reclassified with the IPCC emission factors and climate regions (boreal, temperate, tropical), associated with the three broad land-use types CL, FL, and GL[11]. Classes 11, 14, and 20 were assigned to CL, classes 40, 50, 60, 70, 90, 100, 110, 130, 160, and 170 were assigned to FL, and classes 120, 140, and 150 were assigned to GL. Class 190, associated with mixed urban areas, was assigned to GL. We assigned the mixed class 30 to a combination of the FL, GL and CL emission factors, and class 180 to a combination of FL and GL. Classes 200–230 could not be assigned to any land-use type and were disregarded. In order to improve land classification we combined to this source also CL areas from the EarthStat database[44]. Since this map indicates the probability of the presence of CL between 0 and 1, we discretized it with a threshold of 0.5, considering all areas with higher probability as being CL. We then updated the former land use map by superimposing the CL classification to all these areas.

We combined the resulting peatland land use map with a global climate classification according to Köppen[45], broadly reclassified to the generic climate classes polar+boreal, temperate, and tropical, obtaining a map with $5 \times 3 = 15$ emission classes (Supplementary Fig. 2). These climatic zones are therefore not based on latitude, but on ecological and climatic considerations. We then assigned an emission factor to each class based on ref. [11], after having aggregated the emission factors by climate and land use (polar and boreal climates were assigned the same emission factors) (Supplementary Table 2). We adopted GHG emission factors from ref. [11] by including $CO_2$, $N_2O$, $CH_4$, and dissolved organic carbon DOC in the overall calculation. Given the high uncertainty of the data we decided on a conservative statistical treatment. Uncertainty in the emission factors is considered as uniform probability distribution inside each of the three major classes (CL, FL, and GL), and uncertainty boundaries are therefore expressed as a range. The range considers explicitly maximum and minimum instead of confidence intervals. The resulting map shows the potential peatland distribution and spatially assigns potential GHG emissions from peatland degradation to the potential global peatland area (Fig. 1).

Inside each country we first assumed that all the cropland areas on peatland are fully degrading, and therefore all the CL areas have been assigned land degradation factor 1. We then calculated a degradation ratio (between 0, non-degraded, to 1, fully degraded) using the area of degraded peatland by country reported by ref. [10]. In case that this degradation ratio was indicating a smaller area than what indicated by the cropland areas, we assumed the latter to be the correct estimate. In case, that the relative cropland land use on peatland was smaller than the degradation ratio for a specific country, we subtracted the cropland/peatland ratio from the degradation ratio reported by ref. [10] and applied the remaining degradation ratio uniformly among all the other land use types.

Degradation refers to drainage, the latter being a precondition for further management. We alternatively use both drained and degraded, based on the premise that drainage (i.e., a human activity) is a precondition for management which always induces peat degradation (i.e., a changing ecosystem property) as indicated by changing GHG fluxes (i.e., the measure for degradation). We generated this way a degradation map with values between 0 and 1 expressing the (probabilistic) fraction of degraded peatland in that area (Supplementary Fig. 1).

We multiplied the peatland degradation map (Supplementary Fig. 1) with the potential emission map to calculate a probabilistic map of actual emissions from peatland degradation (Fig. 2). Total emissions are calculated on a per pixel base and then aggregated by climate and land use to produce data in Tab. 1. The peatland area reported by ref. [10], and the cartographic data we assembled are statistically comparable (Spearman's p-value 0.02). The peatland area reported by ref. [10], which contributes to our degradation estimates with the fraction of degraded peatland per country, is distinctly smaller than the cartographic area depicted in Fig. 1 but it is assumed to be representative of the share of degrading peatlands at global scale. Given the generous estimate on which the original map from ref. [4] is based, since it includes in many areas (where data are missing) also histosols and gleysols, we rescaled the peatland areas on a global area estimate based on what is reported in the table from ref. [4], updated with the tropical area reported by ref. [5] plus the area reported by ref. [6].

**Soil carbon to nitrogen ratios**. We used the ISRIC WISE database[23] to extract C/N ratios for mineral soils. We excluded all Histosols, and any soils with C/N ratios

of either <5 or >100 and soil organic C concentration > 200 mg g$^{-1}$, assuming that they carry analytical errors (C/N ratios) or represent organic soils not classified as Histosols. The median C/N ratio of this reduced data set ($n = 22{,}959$) is 10.7.

For organic soils, we took boreal and temperate peat C/N values from ref. [8]. Literature values for C and N contents of tropical peats were compiled to get a generic C/N value for tropical peats (Supplementary Data 1). Samples represent various countries (Malaysia, $n = 14$; Indonesia, $n = 41$; Thailand, $n = 1$; Peru, $n = 1$; Brunei, $n = 12$; Panama, $n = 5$; Micronesia, $n = 2$; Mexico, $n = 1$; Bangladesh, $n = 1$; Democratic Republic of the Congo, $n = 1$; Colombia, $n = 5$; Guyana, $n = 2$; Rwanda, $n = 12$; Papua New Guinea, $n = 12$; Bolivia, $n = 2$), and various land-use types (forest/plantation, $n = 40$; natural, $n = 24$; cropland and rice paddy, $n = 20$; other, incl. grazing, degraded, deforested, and extraction sites, $n = 19$; unknown, $n = 9$).

**Future scenario and mineral soil mitigation potential.** Global GHG emissions in the future projection in Fig. 3 were calculated based on a combination of the updated C stocks from refs. [4–6], with the global peatland degradation ratio reported by ref. [10]. Global peatland distribution came from our map. The mitigation potential of agricultural soils was calculated based on Eq. 1 reported by ref. [31]. We determined maximum and minimum values according to their calibrated values, and then added a prudential uncertainty term corresponding to 1/5 of the value on both ends. Total area of arable land was considered 1563 Gha and for meadow and pasture land was 3360 Gha[32]. The average global soil C content for the calculation was 1.35% and was derived from the ISRIC WISE database[23]. As a result, we obtain a mineral soil sequestration of 24–64 Gt C within 63 years, similar to the range reported by ref. [31] (32–64 Gt over 87 years). We calculated the future scenario considered in this study (increased degrading peatlands in Fig. 3) according to a Michaelis−Menten function describing the increase. The function was set to reach half saturation, i.e., a doubling of the current degraded area within 60 years after present day.

**Data availability.** The data reported in this paper are detailed in the main text and its supplementary information files. Shape files for Figs. 1 and 2, and Supplementary Figs. 1 and 2 are available from L. Menichetti on request.

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

## Acknowledgements

We acknowledge the help of C. Wüst-Galley with ArcGIS and K. Budge for critically reading the manuscript. Shape files delineating organic soil areas were kindly provided by Zicheng Yu (Department of Earth and Environmental Sciences, Lehigh University,

Bethlehem, Pennsylvania, USA), the Swedish Geological Union, the Estonian Land Board (Department of Geology), the Tasmanian Department of Primary Industries, Parks, Water and Environment, Jukka Miettinen (Centre for Remote Imaging, Sensing and Processing CRISP, National University of Singapore), and by Maria Aljes, Northwest German Institute of Forest Research, Göttingen, Germany.

## Author contributions

J.L. developed the idea, compiled tropical peat data, and wrote the text. L.M. wrote the text, compiled boreal and temperate peat data, and performed the calculations.

## Additional information

**Competing interests:** The authors declare no competing interests.

