## [Peer Review File · Nature Communications]

Reviewers' comments:

Reviewer #1 (Remarks to the Author):

The manuscript includes a new, spatially disaggregated assessment of global areas of 'degraded' organic soils. These soils are presently believed to be a significant source of greenhouse gas emissions from the land use sector but that with restoration could serve as an important GHG mitigation activity. By overlaying the peatland map of Yu et al., with the ESA (300m) global land cover map, and normalizing with country-level estimates of 'managed/degraded' organic soils (from Joosten), the authors derived a land-cover X climate stratified data set to which the latest IPCC wetland soils emission factors (which are similarly stratified) could be applied to assess GHG emissions from these organic soils. The new estimates for global total emissions are similar to several previously reported, but the authors suggest this is due to an offsetting effect of a larger area but with lower emission factor values used in their analysis. To my knowledge, this synthesis of landcover, organic soil occurrence, and disturbance frequency is new and would be of interest to many, particularly if the spatial data sets were made available on-line.

However, I think the authors take a serious 'wrong turn' in that their main analysis is structured as a (false) 'choice' to either mitigate emissions from organic soils or from mineral soils! (Line 12: 'we quantify the environmental costs of counterbalancing GHG emissions from drained peatlands by enhanced C storage in mineral soil'). This is particularly strange given that most related work on land use GHG mitigation includes both. I don't see any logic whatsoever that links mitigation on organic soils vs non-organic soils as somehow competing or the one precluding the other. Certainly one can compare different mitigation options (in terms of cost, efficiency, potential, etc) but that's not effectively done in this paper.

It's unfortunate that the authors didn't further develop the question of the mitigation potential of restoring organic soils (e.g., what practices are effective, where could they be implemented, what are the costs, policy options, etc.), which would be a novel and valuable undertaking. Instead they seem to stop at taking the new estimates of total emissions from organic soils, and equating that with mitigation potential, which is not particularly helpful. In lines 40-43 they seem to suggest that maybe it's not possible (or at least it's very difficult) to rehabilitate peatlands after all, which is why they want to look instead at 'compensating' peatland emissions by improving management on non-peatland soils! My suggestion is to first assess the potential for peatland GHG mitigation, independently, assessing technical, economic and/or policy barriers as best you can, at which point comparing to other mitigation options in or out of the land use sector could be done.

Some additional minor comments:

Detailed:

Ln 16 - change 'unless' to 'if'

Ln 37 - probably add directed combustion (with GHG emissions) for energy from mined peat

Ln 74 - Table 1 - denotation of 'mixed pixels' in the land use determination (e.g. CL+GL+FL) should be defined in table text; also suggest placing (vertically) in table after the main LU categories

Ln 87 - Fig. 1 - I don't believe that the map can be depicting the actual areal coverage of managed/disturbed peatlands that are emission sources!! - the colored areas appear contiguous for huge swaths of Canada/Alaska and northern Russia/Finland! Needs a redesign!

Ln 146 - Suggest defining 'degradation' more explicitly here and elsewhere in the paper. There's a huge difference in emissions from 'managed peat soils', from partial drainage on grazed natural vegetation; to intensive drainage, liming, tillage, fertilisation for vegetable production; to mining, drying and combusting. How this range in emission-causing activities were differentiated (or not?) needs to be made clear

Ln 154-158 - Clarify entire paragraph - impossible to make sense of uncertainty assessment

Reviewer #2 (Remarks to the Author):

A. OK

B. Very original and very interesting

C, D. all good

E, F. See my general and specific comments to the authors attached below.

G. As far as I can tell, yes.

H. All clear.

See attached

Reviewer #3 (Remarks to the Author):

Dear Authors & Editors,

A. Summary of key results:

- 1) Mineral soil C sequestration can counterbalance GHG emissions from drained peatlands until a compensation point, reached after 149 (45-758) years; whereafter peatland emissions can no longer be offset.
- 2) Compensating for GHG emissions from drained peatlands via mineral soil C sequestration is area and nitrogen costly.
- 3) Peatland protection is a highly efficient and underappreciated mitigation measure in the land-use sector.

B. As far as I know, such research and claims were not been published yet. They are of interest to others in the community, especially in the field of climate change mitigation in the land use sector and in the field of 'climate smart soils'.

C. Greenhouse gas emissions from drained peatlands/organic soils have been calculated on a global level while using the latest default emission factors from the IPCC wetland supplement, Yu et al. (2010) and Joosten (2010) as organic soil/peatland datasets, and the 2016 European Space Agency Global Land Cover Map. The emissions from drained organic soils/peatlands have been compared to the sequestration potential of mineral soils. The overall approach to compare emissions from drained organic soils with the sequestration potential of mineral soils is valid, but the quality of the datasets for deriving the area of drained organic soils and the application of the land use dataset needs to be improved.

D. Uncertainties in the background dataset for drained organic soils seem to be quite large and should be addressed

E. I think that the necessary data background improvement would make the drawn conclusions more reliable and robust.

F. Unfortunately, the paper is in its current state not convincing for me. My largest concerns are related to the used organic soil/peatland data background from Yu (2010) and Joosten (2010), and the application of the European Space Agency Global Land Cover Map. The latter has been used to partition the global peatland area reported by Joosten (2010) in cropland, grassland and forestry.

- The dataset of 'peatlands' from Yu et al. (2010) has been collated from very different sub-datasets. For Southeast Asia, Central and South America and Africa, data from the Harmonised World Soil Database (HWSD)* has been included. This comprises areas of Histosols (organic soils) and Gleysols (non-organic soils)** - and the vast majority of them as included by Yu et al. (2010) for Central and South America and Africa are Gleysoils and not Histosols.

Especially for Southeast Asia, but also for parts of Africa, Central and South America recent GIS data is or might be available (cf. Barthelmes et al. 2015a; cf. Joosten 2012; Draper et al. 2014). Also for Papua New Guinea sufficient organic soil GIS-data*** can be found (Papua New Guinea is in Figure 1 indicated with 'data not available') and moreover for several peatland-rich, northern European countries (Barthelmes et al. 2015b). I'm aware that the consolidation of these diffuse datasets is very time consuming, but for a publication in Nature Communications I would suggest to compile up-to-date and comprehensive background datasets.

- Joosten (2010) was a first attempt to give an overview on the extent and status of peatlands for all countries of the World based on the Global Peatland Database****. A recent update of this database has been compiled in 2015***** that now has a more satisfactory data background. It might be an option to consider this new dataset on organic soils/peatlands for the paper. Nevertheless, the quality of area data for peatlands and organic soils is for many countries still quite poor, but useful research outcomes are published continuously.

- I wonder if all the ESA-GLCM codes that are merged to 'Forest Land' or 'Grassland' (cf. Table 1) have been regarded drained. There occur many different undisturbed, flooded or waterlogged forests and grasslands on peatlands/organic soils. Especially ESA-GLCM codes 140, 150, 170 and 180 might include pristine peatlands/organic soils. If erroneously classifying them drained would explain why in Figure 1 e.g. the organic soils of Zambia and the Congo Basin are erroneously depicted completely drained with high emissions per area although they are largely pristine in reality. The same applies for the vast areas in Siberia and Canada - they are largely undisturbed but depicted as medium GHG emitting areas in Figure 1. To derive a map like Figure 1, drainage information is needed to distinguish artificially changed from natural peatland forests and grasslands, and the ESA-GLCM seems to be insufficient without this information. Moreover, Figure 1 seems to me generally misleading, because the emission hotspots from drained organic soils/peatlands are, as far as I know, Europe and Southeast Asia and not Africa, South America, Canada and Asian Russia as it looks like in Figure 1. If my considerations are not correct, I would like to know on which basis the peatlands/organic soils of the mentioned regions have been classified 'drained'.

The application of latest emission factors from the IPCC wetland supplement has basically been well done. Only one question: How to handle drained organic soils/peatlands in tropical montane regions that reach out from 1000 to 4-5000 m.a.s.l. (e.g. in the flanks of the Rift Valley and the Andes)? Greenhouse gas measurements are widely lacking for these regions. But GHG emissions are bound to e.g. temperature and the temperature decreases with the altitude. Would it be closer to reality to use temperate emission factors for drained organic soils/peatlands in tropical montane areas?

In Table 1 the area of boreal Forest land appears to be too low for me, but I do not know for what reason...

* <http://webarchive.iiasa.ac.at/Research/LUC/External-World-soil-database/HTML/>

** http://www.isric.org/sites/default/files/major_soils_of_the_world/set4/gl/gleysol.pdf

***<http://www.nari.org.pg/mapping-geographical-information-systems#pgris>

**** <http://www.greifswaldmoor.de/global-peatland-database-en.html>

***** [http://www.greifswaldmoor.de/files/dokumente/15_12_01_Wetlands International \(2015\) Save Peat for Less Heat.pdf](http://www.greifswaldmoor.de/files/dokumente/15_12_01_Wetlands%20International%20(2015)%20Save%20Peat%20for%20Less%20Heat.pdf)

References:

Barthelmes A, Ballhorn U & J Couwenberg (2015a) Consulting Study 5: Practical guidance on locating and delineating peatlands and other organic soils in the tropics. The High Carbon Stock Science Study, available at: <http://www.carbonstockstudy.com/carbonstockstudy/files/6f/6f24a8ba-bd4e-42bb-8848-bedc7db6168d.pdf>

Barthelmes A, Couwenberg J, Risager M, Tegetmeyer C & H Joosten (2015b). Peatlands and Climate in a Ramsar context - A Nordic-Baltic Perspective. Nordic Council of Ministers, Rosendahls-Schultz Grafisk, Copenhagen, Denmark, 81 p. + Annexes.

Draper FC, Roucoux KH, Lawson IT, Mitchard ETA, Coronado ENH, Lähteenoja O, Montenegro LT, Sandoval EV, Zaráte R and TR Baker (2014). The distribution and amount of carbon in the

largest peatland complex in Amazonia. *Environ. Res. Lett.* 9: 1-12.

Joosten, H. The global peatland CO₂ picture. Peatland status and drainage related emissions in all countries of the world. 36 (Wetlands International, Ede, NL, 2010).

Yu ZC, Loisel J, Brosseau D P, Beilman DW & SJ Hunt (2010) Global peatland dynamics since the Last Glacial Maximum. *Geophysical Research Letters* 37, doi:10.1029/2010gl043584.

G. Especially in the field of organic soil and peatland mapping and available GIS-data the paper should address and include more previous work. This will automatically be done if the background data of drained organic soils/peatlands will be improved.

H The paper is written with a satisfactory lucidity and appropriateness in summary, abstract, introduction and conclusions.

I'm convinced that drained organic soils/peatlands provide a huge potential for climate change mitigation due to rewetting, because they are strong GHG emission sources. Furthermore, a bunch of vital ecosystem services can be restored due to rewetting. Discussion and public awareness rising is needed, because drained organic soils/peatlands are preferably used for agriculture and forestry and the opposition against their rewetting is often quite strong. I guess the paper would influence the thinking and intensify the discussions in the fields of 'climate smart soils' and climate change mitigation in the land use sector. I would like to see it published, but this would need from my point of view a major revision, especially regarding the data background.

All the best!

Reviewer 1

The manuscript includes a new, spatially disaggregated assessment of global areas of 'degraded' organic soils. These soils are presently believed to be a significant source of greenhouse gas emissions from the land use sector but that with restoration could serve as an important GHG mitigation activity. By overlaying the peatland map of Yu et al., with the ESA (300m) global land cover map, and normalizing with country-level estimates of 'managed/degraded' organic soils (from Joosten), the authors derived a land-cover X climate stratified data set to which the latest IPCC wetland soils emission factors (which are similarly stratified) could be applied to assess GHG emissions from these organic soils. The new estimates for global total emissions are similar to several previously reported, but the authors suggest this is due to an offsetting effect of a larger area but with lower emission factor values used in their analysis. To my knowledge, this synthesis of landcover, organic soil occurrence, and disturbance frequency is new and would be of interest to many, particularly if the spatial data sets were made available on-line.

However, I think the authors take a serious 'wrong turn' in that their main analysis is structured as a (false) 'choice' to either mitigate emissions from organic soils or from mineral soils! (Line 12: 'we quantify the environmental costs of counterbalancing GHG emissions from drained peatlands by enhanced C storage in mineral soil'). This is particularly strange given that most related work on land use GHG mitigation includes both. I don't see any logic whatsoever that links mitigation on organic soils vs non-organic soils as somehow competing or the one precluding the other. Certainly one can compare different mitigation options (in terms of cost, efficiency, potential, etc) but that's not effectively done in this paper.

Authors. We are grateful for this comment as it shows that we did not explain our goals in a proper way and used a partially misleading wording. We do not aim to suggest compensating organic soil emissions by mineral soil C sequestration. Rather, we exemplify how much more environmental cost and effort it would need to take advantage of the mineral soil option in comparison to the organic soil option, particularly because of the nitrogen and area involved. This does not mean, that mineral soil options are worth- or meaningless, but that priority should be given for the low-hanging fruit. We revised substantially and clarified that issue at several points in the text.

It's unfortunate that the authors didn't further develop the question of the mitigation potential of restoring organic soils (e.g., what practices are effective, where could they be implemented, what are the costs, policy options, etc.), which would be a novel and valuable undertaking. Instead they seem to stop at taking the new estimates of total emissions from organic soils, and equating that with mitigation potential, which is not particularly helpful. In lines 40-43 they seem to suggest that maybe it's not possible (or at least it's very difficult) to rehabilitate peatlands after all, which is why they want to look instead at 'compensating' peatland emissions by improving management on non-peatland soils! My suggestion is to first assess the potential for peatland GHG mitigation, independently, assessing technical, economic and/or policy barriers as best you can, at which point comparing to other mitigation options in or out of the land use sector could be done.

Authors. Also in response to the previous comment of the reviewer, we clarify that our MS is not intended to address the technical aspects of mitigation strategies in organic soils. While the ecological conditions for such rehabilitation is relatively well known, the economic and social dimension of taking intensively used organic soils out of management is far less known and far beyond the scope of our work. In the revised version, we recon rehabilitation as well as paludiculture as feasible ways towards GHG-neutral organic soils and added the respective, mostly very recent, references.

Some additional minor comments:

Authors. Detailed:

Ln 16 - change 'unless' to 'if'
Authors. Changed

Ln 37 - probably add directed combustion (with GHG emissions) for energy from mined peat
Authors. Added

Ln 74 - Table 1 - denotation of 'mixed pixels' in the land use determination (e.g. CL+GL+FL) should be defined in table text; also suggest placing (vertically) in table after the main LU categories
Authors. Denotation has been added to table text.

Ln 87 - Fig. 1 - I don't believe that the map can be depicting the actual areal coverage of managed/disturbed peatlands that are emission sources!! - the colored areas appear contiguous for huge swaths of Canada/Alaska and northern Russia/Finland! Needs a redesign!

Authors. The map has been redesigned, and the emissions colors were due to an error in the map interpretation which previously depicted potential emissions (this has been noticed also by another referee). Fig. 1 presents now two different maps, potential and estimated emissions, and the corresponding explanation.

Ln 146 - Suggest defining 'degradation' more explicitly here and elsewhere in the paper. There's a huge difference in emissions from 'managed peat soils', from partial drainage on grazed natural vegetation; to intensive drainage, liming, tillage, fertilisation for vegetable production; to mining, drying and combusting. How this range in emission-causing activities were differentiated (or not?) needs to be made clear.

Authors. As stated in our MS, we refer to GHG emissions from drainage and biological oxidation, not peat fires/combustion. Degradation refers to drainage, a precondition for further management. We use both terms, based on the premise that drainage (i.e., a human activity) is a precondition for management which induces peat degradation (i.e., a changing ecosystem property) as indicated by changing GHG fluxes (and other soil parameters such as subsidence that we do not refer to in our study) (i.e., the measure of degradation). We more explicitly refer to this terminology in the methods section now.

The reviewer correctly argues that emissions differ depending on the factors listed above. It is impossible, at the global scale, to infer the type and intensity of management at a level more disaggregated than land-use. We revised the uncertainty calculations, and we are now considering the full range of values in the IPCC wetland supplement (<http://www.ipcc-nggip.iges.or.jp/public/wetlands/>) without assuming any probability distribution for them. We report the mean for convenience, but the uncertainty range is now calculated assuming uniform distribution and is therefore as cautious as it can be.

We believe that our estimate, although highly uncertain, represents the best detail achievable nowadays with the information available. It can therefore represent both a useful tool at a global scale and a stimulus to improve the availability of information.

Ln 154-158 - Clarify entire paragraph - impossible to make sense of uncertainty assessment

Authors. Upon revision, the uncertainty assessment has been re-done on more conservative principles (assuming uniform distributions and therefore a range). This also greatly simplified the procedure for its calculation, which is hopefully clear now.

Reviewer 2

General comments

This is a very nice, very straight forward study. Congratulations! I made a few minor comments while reading the MS, which I am listing below. There is only one major comment/remark: I guess it is fair to compare losses of carbon (C) from peatland degradation

or conversion with potential gains thru C-sequestration in agricultural, mineral soils. Such comparison leaves aside all other sources of GHG emissions (e.g. methane from peatlands or paddy soils or nitrous oxide emissions from upland soils). But, it sounds inconsistent to me when the authors then argue that “additional N₂O emissions from mineral soil may offset a substantial fraction of the newly sequestered C in terms of the net warming effect...”. I would assume that if such additional emissions are taken into the debate, so should also methane emissions from wetlands/peatlands. Would be good if the authors could address this issue.

Authors. We clarify that, for our emission estimate of drained peatlands, not only CO₂, but also N₂O, CH₄ and DOC are taken into account. That is why all emissions in Table 1 are given in units CO₂-equivalents. We specify that information in the methods section.

On the other hand, the authors may want to add one sentence underlining the ecological/hydrological importance of peatlands as well as the fact that these ecosystems often comprise quite a unique and rich biodiversity.

Authors. We refer to the functioning of peatlands beyond the GHG issue in the introduction now.

Specific comments

Line 57: probably better to write “... (= 0.33 Gt C; range 0.10–0.59 Gt C)...”.

Authors. We did not change the way we express ranges in the text and leave it to the editor to decide.

Line 58-59: “net biogenic terrestrial CO₂ sink of –5.3 (±4.5) Gt CO₂” – a bit unclear: is this negative sink thus a source? If so, then better remove the minus sign and call the whole thing what it is, a source.

Authors. The sink is a sink (the number refers to the terrestrial biosphere, not only to soil, see cited reference). We use the negative sign following i) the original publication (Tian et al. 2016, Table 1) and ii) our own nomenclature where we always assign positive values to sources.

Table 1 and methodology chapter: maybe the land use classes could be spelled out, e.g. ‘crop’, ‘grass’, ‘forest’.

Authors. Abbreviations for land use classes are spelled out in the Table footnote and in the methods section.

Line 177-178: “Future scenario and mineral soil mitigation potential - The mitigation potential of agricultural soils was calculated according to Eq. 1 reported by Sommer and Bossio” – well, if only equation 1 of this publication was used, then the authors adopted only half of the methodology presented by Sommer and Bossio (2014), as on top of the SOC sequestration dynamics, they added an area-adoption scenario. The underlying assumption is that not all land can be converted into a C-sink immediately, but adoption of C-sequestration practices takes time. I do not assume that this is a major flaw and that the basic message will remain the same, irrespectively; yet, just to flag this issue.

Authors. The reviewer is right - the adoption of only equation 1 from Sommer and Bossio (2014) is done in order to obtain the most optimistic scenario (an instant adoption of mitigation measurements on all the global agricultural area). This choice does not, however, influence the calculation of our equivalence points, since these are solely based on the total mitigation potential of agricultural soils.

Reviewer 3

A. Summary of key results:

- 1) Mineral soil C sequestration can counterbalance GHG emissions from drained peatlands until a compensation point, reached after 149 (45-758) years; whereafter peatland emissions can no longer be offset.
- 2) Compensating for GHG emissions from drained peatlands via mineral soil C sequestration is area and nitrogen costly.
- 3) Peatland protection is a highly efficient and underappreciated mitigation measure in the land-use sector.

B. As far as I know, such research and claims were not been published yet. They are of interest to others in the community, especially in the field of climate change mitigation in the land use sector and in the field of 'climate smart soils'.

C. Greenhouse gas emissions from drained peatlands/organic soils have been calculated on a global level while using the latest default emission factors from the IPCC wetland supplement, Yu et al. (2010) and Joosten (2010) as organic soil/peatland datasets, and the 2016 European Space Agency Global Land Cover Map. The emissions from drained organic soils/peatlands have been compared to the sequestration potential of mineral soils. The overall approach to compare emissions from drained organic soils with the sequestration potential of mineral soils is valid, but the quality of the datasets for deriving the area of drained organic soils and the application of the land use dataset needs to be improved.

Authors. The referee has undoubtedly a good point, and the data uncertainty is at the global level quite relevant. We worked hard to incorporate all the improvements of the past six years in our new potential peatland dataset (see also following comments for details), consulting several sources and we collected more information in order to present the state-of-the-art of global knowledge about peatland extension. Given our purpose (calculating a representative land use proportion), we also chose for developing our updated peatland map an inclusive approach, and the map is privileging protection from false negatives (it is therefore on the upper side of the uncertainty range).

There is no finally valid way to address this concern, meaning that at the global scale there will still be several uncertainties and there is no consensus among different sources with respect to the area and allocation of peatlands. Understanding that the global peatland extension is a matter of choices in the criteria utilized (e.g., which soil classes to incorporate, carbon content threshold etc.), and that there is no true answer at this point, estimates will tend to either underestimating or overestimating the global picture.

D. Uncertainties in the background dataset for drained organic soils seem to be quite large and should be addressed.

Authors. The uncertainties are in this respect indeed quite large, but this is due to the state of knowledge at the global level. The estimates collected by Joosten, 2010 are still to date the best available source, but also given the uncertainty in the estimate of total peatland area and from the choices needed in this sense, their uncertainty is also hard to be evaluated systematically.

Our study represents, nevertheless, both a review and compilation of the globally available information and a perspective on the role of peatlands in climate change mitigation. While the latter objective is met even within the high uncertainty surrounding the available data, the former reflects the actual state of knowledge and can (and hopefully will) be updated in the future years. We believe that both of the objectives of this study will contribute to the advancement of science in this direction.

E. I think that the necessary data background improvement would make the drawn conclusions more reliable and robust.

Authors. We improved the data, partially as suggested and partially by developing new approaches. We want to thank the referee to point out this flaw and to push us to do better.

F. Unfortunately, the paper is in its current state not convincing for me. My largest concerns are related to the used organic soil/peatland data background from Yu (2010) and Joosten (2010), and the application of the European Space Agency Global Land Cover Map. The latter has been used to partition the global peatland area reported by Joosten (2010) in cropland, grassland and forestry.

- The dataset of 'peatlands' from Yu et al. (2010) has been collated from very different sub-datasets. For Southeast Asia, Central and South America and Africa, data from the Harmonised World Soil Database (HWSD) has been included. This comprises areas of Histosols (organic soils) and Gleysols (non-organic soils)** - and the vast majority of them as included by Yu et al. (2010) for Central and South America and Africa are Gleysoils and not Histosols.*

*Especially for Southeast Asia, but also for parts of Africa, Central and South America recent GIS data is or might be available (cf. Barthelmes et al. 2015a; cf. Joosten 2012; Draper et al. 2014). Also for Papua New Guinea sufficient organic soil GIS-data*** can be found (Papua New Guinea is in Figure 1 indicated with 'data not available') and moreover for several peatland-rich, northern European countries (Barthelmes et al. 2015b). I'm aware that the consolidation of these diffuse datasets is very time consuming, but for a publication in Nature Communications I would suggest to compile up-to-date and comprehensive background datasets.*

*- Joosten (2010) was a first attempt to give an overview on the extent and status of peatlands for all countries of the World based on the Global Peatland Database****. A recent update of this database has been compiled in 2015***** that now has a more satisfactory data background. It might be an option to consider this new dataset on organic soils/peatlands for the paper. Nevertheless, the quality of area data for peatlands and organic soils is for many countries still quite poor, but useful research outcomes are published continuously.*

Authors. The suggested update has been considered We integrated our dataset with other sources, as detailed below ('Regarding peat area and emission estimate consolidation').

- I wonder if all the ESA-GLCM codes that are merged to 'Forest Land' or 'Grassland' (cf. Table 1) have been regarded drained. There occur many different undisturbed, flooded or waterlogged forests and grasslands on peatlands/organic soils. Especially ESA-GLCM codes 140, 150, 170 and 180 might include pristine peatlands/organic soils. If erroneously classifying them drained would explain why in Figure 1 e.g. the organic soils of Zambia and the Congo Basin are erroneously depicted completely drained with high emissions per area although they are largely pristine in reality. The same applies for the vast areas in Siberia and Canada - they are largely undisturbed but depicted as medium GHG emitting areas in Figure 1. To derive a map like Figure 1, drainage information is needed to distinguish artificially changed from natural peatland forests and grasslands, and the ESA-GLCM seems to be insufficient without this information. Moreover, Figure 1 seems to me generally misleading, because the emission hotspots from drained organic soils/peatlands are, as far as I know, Europe and Southeast Asia and not Africa, South America, Canada and Asian Russia as it looks like in Figure 1. If my considerations are not correct, I would like to know on which basis the peatlands/organic soils of the mentioned regions have been classified 'drained'.

Authors. The localization of the "drained" status in our study is probabilistic. The data are derived by assuming an uniform distribution of the degraded areas among all the land uses, and its purpose is to offer an aggregated estimate (Table 1) as precise as possible of the global relevance of peatland emissions. We clarified our approach in the methods section now in more detail, see also below. Nevertheless Figure 1 in the former revision of the manuscript was misleading, since it was not explained well enough. The figure has now been revised.

Authors 1 Regarding peat area and emission estimate consolidation:

The global data available are indeed not always the best in quality, particularly for tropical peatlands. We nevertheless could work on several improvements, partly following the referee's suggestions and partly developing own new approaches.

One thing that must be noticed, and that it is now well specified in the manuscript, is that our map should be considered an effort to be on the safe side and not to neglect areas where peatlands are known to exist in relevant amounts but for which no official data have yet been produced (examples: Africa and South America).

For our potential peat extension map (Fig. 1a,b in revised MS) we selected an approach robust toward false negatives. We selected this approach to get a representative area for calculating the distribution of peatlands among land use classes (but not the actual emissions!). This approach increases the accuracy of land-use assignments, and offers a map that does not neglect any important areas.

However, for the estimation of the emissions we decided to be conservative and to utilize the area data from Joosten, 2010. Joosten's data is based on verified sources and therefore likely underestimates the actual peatland area. At the same time these data are more resistant toward false positives, and we preferred this in order to offer a robust estimate of the (still highly uncertain) emissions. Please refer to the map of the sources (incomplete, reporting only the major ones) below for a graphical representation of the uncertainty surrounding global peatland area.

Map shows the spatial distribution of peatlands according to three different sources. Note the pronounced deviation among sources for almost all continents.

The correspondence between our new map and the area from Joosten (2010) is highly significant, and a linear regression on the peatland area by country from the two sources reveals an adjusted R² of 0.976. Still we preferred to rely on the more robust and conservative source for the estimation of GHG emissions, providing the map for a more accurate land-use determination and incorporating subsequent updates while the global knowledge increases.

As a reference for checking the different approaches of peatland identification we usually referred to the peatland areas we received from the respective geological services (Sweden, Tasmania, Estonia), which we regard as reliable. **We then worked on the following nine improvements:**

1) The utilization of the SoilGrid data, as suggested in Barthelmes et al. (2015), can help in some cases but is limited for organic soils. Its Histosol area is clearly neglecting some important peatlands, since this area is only 3.1 million km² and even the area reported by

Joosten, much likely an underestimation already, is 3.8 million km². The baseline for this dataset has been developed starting from a data base sharing most of the points with the HWSD, but with process-based geostatistic techniques by integrating several datasets, and the overall level of information increased. The additional data layers did not include, nevertheless, a water table data.

This makes the SoilGrid data, particularly the soil C layer, well suited for non-extreme soils where soil zonality principles apply but unable to catch the processes leading to the formation of organic soils. The resulting maps of histosols can sometimes reflect pretty well the peatland areas, but in some other cases they miss it completely (e.g. in the Congo or Amazon basin, or in Tasmania).

We confronted also the results of two thresholds applied to the soilgrid C content data (12% and 20% carbon) with a peatland map that we consider as accurate (the Swedish peat map, which we received from the Swedish Geological Union), and also in this area the SoilGrid data showed quite some inconsistencies. This dataset has been therefore used in conjunction with others only on a case-by-case criterium, but is not considered as representative for organic soils globally.

2) Another possibility for validating the peatland maps available is through wetland maps. For this purpose, we first tested the UNEP map from 1993, which assembles several sources (Dugan, 1993). Again, in several cases, the known peatland areas and wetlands overlap, but not in all cases, leading both to false negative and positive results. The dataset cannot be trusted completely. The area of properly marked peatlands in this dataset is only 0.25 million km², and including freshwater marshes increases this to 0.5 million km² but clearly causes many false positives.

We also integrated this dataset with another one from wetland modeling, based on an approach that privileged false positives to be on the safe side (Zhu and Gong, 2014). We filtered this dataset by considering only areas with >50% of the time in wet conditions, and we added this area to the UNEP wetland maps to compile a prudential map of possible wetland areas. Also with this approach, when tested over the known peatland maps, we revealed several inconsistencies and both false negatives and false positives (even if bearing in mind that a wetland must not be a peatland). We therefore used this map in conjunction with the SoilGrid C content data only to manually check country by country to spot possible problems, that we then verified through literature analysis.

3) Tasmania

Tasmanian peatlands are now from data received directly from the Tasmanian Department of Primary Industries, Parks, Water and Environment; project TASVEG (Department of Primary Industries and Water, 2009. TASVEG Version 2. Released February 2009. Tasmanian Vegetation Monitoring and Mapping Program, Resource Management and Conservation Division, Hobart). This map is at date the best available source for peatland distribution in the area.

4) Ramsar database

We integrated our potential peatland map by adding the area considered by the Ramsar convention (as suggested by the third reviewer), adding several areas previously neglected by Yu et al. (2010).

5) Estonia

Data for Estonia are now from a dataset from the Estonian Land Board, Department of Geology, mapping the peat deposits.

6) Malaysia, Sumatra and Borneo

Data for this important area are now updated with the map produced by Miettinen et al. (2016).

7) Kyrgyzstan

Data from Alyes et al., (2016), have been included in our map.

8) Information on tropical peatlands are in several cases scarce and not reliable. The decision of leaving the Gleysols as previously proposed by Yu et al. (2010) in the tropical peatland areas is motivated by the area of the Congo basin. This area is in fact known to host a relevant amount of peat. This area is in most cases classified as Gleysols in the FAO world soil map, and often as Ferralsols or Gleysols in the SoilGrid, and therefore basing the classification on histosols alone would miss this important peat area (which is confirmed by the wetland map and partially by the SoilGrid C map too). Similar issues are present in the Amazon basin.

These issues derive again from the azonal characteristics of histosols, which makes them often misrepresented in traditional soil maps.

9) Papua New Guinea data were already included in the calculations, but due to a mistake, they did not show in the map. The area is now included, and it comes from Yu et al., 2010. This particular case is also mostly in agreement with the SoilGrid carbon database.

9) Joosten 2015 does not seem to contain substantial updates from 2010. Still we considered that publication upon revision.

Authors 2 Regarding the map of emissions:

The referee noted an important issue in the previous version of the manuscript, since the map was depicting only the potential emissions (before the calculations, which were already done with the current peatland degradation status). This map has been kept (Fig. 1 a) since we believe it can be useful for the reader, but a second map (Fig. 1 b) is now shown that has been recalculated according to a normalization based on the most recent degradation data reported by Joosten (2010), by country and then applied to our georeferenced database.

The application of latest emission factors from the IPCC wetland supplement has basically been well done. Only one question: How to handle drained organic soils/peatlands in tropical montane regions that reach out from 1000 to 4-5000 m.a.s.l. (e.g. in the flanks of the Rift Valley and the Andes)? Greenhouse gas measurements are widely lacking for these regions. But GHG emissions are bound to e.g. temperature and the temperature decreases with the altitude. Would it be closer to reality to use temperate emission factors for drained organic soils/peatlands in tropical montane areas?

Authors. There is indeed, according to our knowledge, not a single study where GHG emissions from high elevation peatlands in the tropical zone were measured. Tropical mountain peatlands occur in many countries, with varying degrees of utilization. In New Guinea, peatlands of the montane zone (1000 – 2800 m asl) are often drained and managed, whereas subalpine peatlands (2700 – 4550 m asl) are, if at all, only extensively used, e.g. for hunting (Hope 2014). The latter type comprises 3707 out of a total of 5965 km² mountain peatlands in New Guinea, indicating that these higher elevation, cold peatlands do not contribute much to GHG emission from drained organic soils. Mountain peatlands in the Colombian Cordillera Oriental at ca. 3600 m asl, are increasingly drained and utilized since the 1950ies (Benavides 2014). Their degree of disturbance is, however, still low and drained sites still accumulate C. It is therefore not likely, that these ecosystems contribute much to the overall GHG emission drained peatlands in the tropics. Intensively grazed peatlands in the Bolivian Andes (4500 – 4600 m asl) were studied by Hribljan et al. (2014). They found age-depth relationships that still resemble relatively undisturbed, i.e., peat accumulating conditions at two sites. Together, these few available studies indicate that high elevation tropical peatlands exert less disturbance than their lowland counterparts. More importantly,

the vast majority (89 % of all counted pixels) of tropical peatlands (between 23.5 ° N and 23.5 °S) is situated below 400 m asl (Fig. below). Hence, the reported IPCC emission factors do reflect the actual situation in the tropics and cannot be considered biased, notwithstanding that measurements for high elevations would be highly valuable.

Global distribution of tropical peatlands along elevation.

In Table 1 the area of boreal Forest land appears to be too low for me, but I do not know for what reason...

Authors. Tab. 1 shows only the area of forest on peatland areas (and most areas in this class are now, after the review, moved in the mixed class due to the amendment of an error, so the area is now even smaller), not the total forested areas.

G. Especially in the field of organic soil and peatland mapping and available GIS-data the paper should address and include more previous work. This will automatically be done if the background data of drained organic soils/peatlands will be improved.

Authors. Several data sources have been consulted and were incorporated upon revision. The manuscript now reflects the state of knowledge to date.

H The paper is written with a satisfactory lucidity and appropriateness in summary, abstract, introduction and conclusions.

I'm convinced that drained organic soils/peatlands provide a huge potential for climate change mitigation due to rewetting, because they are strong GHG emission sources. Furthermore, a bunch of vital ecosystem services can be restored due to rewetting. Discussion and public awareness rising is needed, because drained organic soils/peatlands are preferably used for agriculture and forestry and the opposition against their rewetting is often quite strong. I guess the paper would influence the thinking and intensify the discussions in the fields of 'climate smart soils' and climate change mitigation in the land use sector. I would like to see it published, but this would need from my point of view a major revision, especially regarding the data background.

References

- Alyes, M., Heinkicke, T. And Teitz, J. 2016. Peatland ecosystems in Kyrgyzstan: Distribution, peat characteristics and a preliminary assessment of carbon storage. *Catena*, 2016, **144**: 56-64.
- Barthelmes A, Ballhorn U & J Couwenberg 2015. Consulting Study 5: Practical guidance on locating and delineating peatlands and other organic soils in the tropics. The High Carbon Stock Science Study, available at: <http://www.carbonstockstudy.com/carbonstockstudy/files/6f/6f24a8ba-bd4e-42bb-8848-bedc7db6168d.pdf>

- Benavides JC. 2014. The effect of drainage on organic matter accumulation and plant communities of high-altitude peatlands in the Colombian tropical Andes. *Mires and Peat* 2014, **15**: 1-1.
- Dugan, P. J. (Ed.). 1993. Wetlands under threat. Mitchell Beazley, London.
- Hope GS. Peat in the mountains of New Guinea. *Mires and Peat* 2014, **15**: 13-13.
- Hribljan JA, Cooper DJ, Sueltenfuss J, Wolf EC, Heckman KA, Lilleskov EA, *et al.* 2014. Carbon storage and long-term rate of accumulation in high-altitude Andean peatlands of Bolivia. *Mires and Peat* **15**: 12-12.
- Joosten H. 2010. The global peatland CO₂ picture. Peatland status and drainage related emissions in all countries of the world. (eds. Wetlands International).
- Miettinen J, Shi C, Liew SC. 2016. Land cover distribution in the peatlands of Peninsular Malaysia, Sumatra and Borneo in 2015 with changes since 1990. *Global Ecology and Conservation*, **6**: 67–78.
- Zhu P, Gong P. 2014. Suitability mapping of global wetland areas and validation with remotely sensed data. *Science China Earth Sciences*, **57**: 2283–2292.
- Yu ZC, Loisel J, Brosseau DP, Beilman DW, Hunt SJ. 2010. Global peatland dynamics since the Last Glacial Maximum. *Geophysical Research Letters* 37, article L13402.

Reviewers' comments:

Reviewer #1 (Remarks to the Author):

In the previous review, my concerns were mainly with what seemed to be a misdirected – or at least unproductive – comparison of GHG mitigation through rewetting of drained (i.e., managed) organic soils with mitigation practices and potentials on mineral (non-organic soils). I think the revised ms is better in that respect. It still retains as a fairly major focus the comparison with mitigation potential of mineral soils, but at least it doesn't read as if they are somehow 'offsetting' or competing objectives. Whether the comparison, as it now stands, is a particularly useful or illuminating one can be argued – to be honest, issues with preserving undrained peatlands as a mitigation option has more in common with avoided deforestation (REDD), and rewetting of organic soils used for annual crops and production forestry, which usually necessitates changes in land use and land cover, are more analogous to mitigation options like afforestation or prairie restoration. However, to the extent that comparison of the relative magnitude of wetland soil mitigation potential to another landuse related topic that has been discussed in the literature (i.e. mineral soil C sequestration!) is a goal of the paper – it might provide a useful reference point.

However, in reviewing the revised version and in trying to resolved some questions that led me to a closer examination of the key foundational papers used in constructing the new database, I have some major concerns with some of the data and the analysis.

First, is the very large discrepancy in the total peatland areas in the new map (944 Mha), with that in the main foundational papers by Yu et al. (440 Mha) and by Joosten (380 Mha). The latter two actually agree relatively well!, also considering that the total of Yu et al. likely contains some area of non-organic soils (Gleysols) as discussed in the response to Reviewer 3. I certainly understand the difficulty in combining multiple data sources into the new map/database, but the more than 2-fold increase in area above that of Joosten needs to be better explained (double-checked??)!! The development of a new – and presumably more detailed and accurate database of global wetland organic soils would be, in my opinion, the most significant outcome of the submitted work and therefor you need to get it right! I don't believe the updating for the individual countries that are listed in the 'Methods' section (starting on line 167) can account for this. Further, looking at the Ramar sites referenced, it's not clear at all how this data is used, as the database consists of wetland 'site' descriptions. They don't appear to be contiguously mapped but there are areas reported for the different sites. However, if you look in detail at a particular site, that area can include open water (lakes, ponds) and an array of various 'wetland type' of systems, but I couldn't find any data that corresponded to soil type. In any case, the present ms is lacking adequate documentation and an explanation for the much larger land base of peatland soils that is derived.

Secondly, the overlay and then classification of peatland into various land use categories is not well done, but I believe could be improved. For example, in the Boreal climate zone, the largest land use category (over 25% of the global total!) is the 'mosiac' of combined cropland, grassland, forest. Even if much of the ESA-GCLM product in this area showed as code 30, it is clear that the area of what would be IPCC classified cropland is vanishingly small (maybe a bit of potatoes in northern Norway, some cabbages along the Peace River in Alberta ?!). That area has to be almost entirely a mixture of forest and 'grassland' bogs – I think you could refine and improve this classification using something like Ramankutty's cropland land covers or even some regional agricultural statistics, to as much as possible differentiate between cropland and non-cropland peat areas.

Along the same lines, the way peat degradation is assigned to land covers is poorly conceived and biases the results. Line 195 states '... based on the assumption that peatland degradation areas are randomly distributed among all the land use classes'. This impacts the results in Fig. 1 and Table 1. I would submit that you should assume that 100% of the cropland land area that is on peat soils, falls into the degraded peatland category – by definition! Growing annual crops on organic soils is the dominant degradation cause (excepting peat mining for energy) – i.e., you have to drain, lime, cultivate, etc (cranberry bogs aside). Even rice on organic soils – likely the only cropland use that doesn't involve complete drainage – would lead to net losses of the organic matter. So the logical approach would be to first assign degraded organic soil area to cropland, and then perhaps assume an equal distribution of remaining degraded peatland area allocated between the grassland and forestland categories, unless other information indicates what systems are most likely to have had peatland drainage.

Accordingly there seem to be lots of problems with Table 1 – e.g., as noted above, for CL in Boreal and Tropical areas, most of the peat area under CL is shown as NOT degrading!; also in the CL Temperate area there is no (0.00) land area shown as only 'CL' yet there is a degrading peat C stock assigned to that category along with calculated GHG emissions; similarly, the tropical mosaic (CL+GL+FL) has no (0.00) area, but also a large degrading peat stock and GHG emissions.

The way in which the degrading peatland area is assigned to land uses, will affect the relative weighting of the emission factors used (which differ by land use and climate) and hence the overall estimates of the GHG emissions. At present it difficult to have confidence in those estimates – and to evaluate the other secondary results that depend on the land cover/peatland allocations, given the outstanding methodological issues.

Reviewer #3 (Remarks to the Author):

Dear authors and editor,

I'm pleased by the changes in the manuscript regarding my main criticism about the peatland dataset. I'm impressed by the extensive methodological explanations and considerations provided from the authors in their response to the first manuscript review.

The methodology is now well explained; the peatland dataset is improved and especially figure 1 has been successfully enhanced.

The authors now provide reasonable and substantiated global estimates on peatland emissions, both, the current and the possible future emissions.

I recommend this manuscript for publishing in Nature Communications, because it is enhanced to appropriate methodological quality and appreciates the very important (and at global level underdeveloped) area of peatland research in context of climate change.

You may give my full name in the online documentation of the review process.

Author's response in *italics*

Reviewers' comments:

Reviewer #1 (Remarks to the Author):

In the previous review, my concerns were mainly with what seemed to be a misdirected – or at least unproductive – comparison of GHG mitigation through rewetting of drained (i.e., managed) organic soils with mitigation practices and potentials on mineral (non-organic soils). I think the revised ms is better in that respect. It still retains as a fairly major focus the comparison with mitigation potential of mineral soils, but at least it doesn't read as if they are somehow 'offsetting' or competing objectives. Whether the comparison, as it now stands, is a particularly useful or illuminating one can be argued – to be honest, issues with preserving undrained peatlands as a mitigation option has more in common with avoided deforestation (REDD), and rewetting of organic soils used for annual crops and production forestry, which usually necessitates changes in land use and land cover, are more analogous to mitigation options like afforestation or prairie restoration. However, to the extent that comparison of the relative magnitude of wetland soil mitigation potential to another landuse related topic that has been discussed in the literature (i.e. mineral soil C sequestration!) is a goal of the paper – it might provide a useful reference point.

We appreciate that the reviewer's and our viewpoint are now converging. In addition to that what has been said above, we underpin that two important aspects of our study, the nitrogen and area demands, which make a difference between mineral and organic soil mitigation measures, have not been addressed explicitly previously.

However, in reviewing the revised version and in trying to resolved some questions that led me to a closer examination of the key foundational papers used in constructing the new database, I have some major concerns with some of the data and the analysis.

We are grateful for the constructive feedbacks and believe that we could tackle all points raised by the referee.

First, is the very large discrepancy in the total peatland areas in the new map (944 Mha), with that in the main foundational papers by Yu et al. (440 Mha) and by Joosten (380 Mha). The latter two actually agree relatively well!, also considering that the total of Yu et al. likely contains some area of non-organic soils (Gleysols) as discussed in the response to Reviewer 3. I certainly understand the difficulty in combining multiple data sources into the new map/database, but the more than 2-fold increase in area above that of Joosten needs to be better explained (double-checked??)!! The development of a new – and presumably more detailed and accurate database of global wetland organic soils would be, in my opinion, the most significant outcome of the submitted work and therefor you need to get it right! I don't believe the updating for the individual countries that are listed in the 'Methods'

section (starting on line 167) can account for this.

First, we stress that the development of a new database on global organic soils is not intended the most significant outcome of our work, but an (important) tool that we need to make a useful comparison of the conditions of different mitigation measures. Our intention, already declared in the manuscript, is to offer an upper limit for the estimate in order to set the range (where former estimate cited by the referee represent the lower limit) where to direct following explorations. Second, we ask the reviewer and the editor to note that the upper limit area has been used to depict the allocation of organic soil GHG emissions and their kinetic, whereas the maximum size of the emissions is still based on the total C stocks in peatland which comes from former estimates (Yu et al. and Dargie et al.).

Detailed response: The referee seems to misunderstand the data reported in the paper by Yu et al. (and we can relate to this since the same happened also to us in the very beginning). In their paper Yu et al. (2010) write: "Data sources for the peatland map (Figure 1) were based on the most up-to-date information available from individual countries or regions in major peatland regions of the world [...]. Some of these peatland data sets are available in shapefile or raster digital formats [...]. For other regions, we mapped peatlands either as histosols and/or gleysols layers as in the Harmonized World Soil Database V1.1 or from digitized paper sources. The peatland areas we used in the peat C pool calculations were derived from the literature [...], rather than directly from the new peatland map presented." The authors decided in that paper to be conservative in their calculations and base them on former estimates (the same on which Joosten built upon), even if their shapefile represents a much bigger area (which is approximately twice of what formerly reported in the literature. We started assembling our dataset from the same shapefile and we measured it).

We took a different choice, and decided instead to update Yu et al. (2010) area estimates with new data, since it appears clear nowadays that former estimates tend to underestimate the global peatland area substantially. By adding histosols to the map we could consider also tropical peatlands (Congo and Amazonas are the most important ones), and this is to date the only way in our opinion to consider these really important areas (see for example Dargie et al., 2017, for some glimpses about new insight on Congo basin peatland areas, that is starting now to be recognized and measured). Since a large peatland area has been confirmed in Congo, the risk of neglecting other important tropical peatlands only because not officially mapped appears now too big and it is clear that the old estimates on global peatland area are underestimates.

Where possible we always completed/substituted the map from Yu et al. (2010) with updated detailed information, such as in the south-east Asia, but when this was not possible we decided that the error we would have committed by using rough estimates (e.g. histosols) would have been smaller than the error we would commit by using a conservative estimate based only on recorded peatland areas (such as former estimates, around 400-500 Mha).

We also believe that it is important that we, as scientific community, start to realize that the 400-500 Mha estimate is insufficient and imprecise, and we admit the lack of

knowledge on global peatland extension. Otherwise we will keep basing our calculations on extremely conservative estimates which neglect a huge peatland area only because of former literature. Our estimate is most probably overestimating the area, but this happens in a literature context where other estimates are badly underestimating it. We believe that this perspective is needed in nowadays literature also to point out the need for information we face, and that the picture will soon become more precise.

N.B: the global map projection has been changed, in order to have a more area-neutral projection and make the new pixel-based calculation possible (Goode-homolosine now, see following answers). The depicted area of peatland therefore slightly changed from previously 944 to 955 Mha.

Dargie, G. C., Lewis, S. L., Lawson, I. T., Mitchard, E. T. A., Page, S. E., Bocko, Y. E., & Ifo, S. A. (2017). Age, extent and carbon storage of the central Congo Basin peatland complex. *Nature*. <http://doi.org/10.1038/nature21048>

Further, looking at the Ramsar sites referenced, it's not clear at all how this data is used, as the database consists of wetland 'site' descriptions. They don't appear to be contiguously mapped but there are areas reported for the different sites. However, if you look in detail at a particular site, that area can include open water (lakes, ponds) and an array of various 'wetland type' of systems, but I couldn't find any data that corresponded to soil type.

The Ramsar sites weight for less than 1 % of the total area we estimated, so any error associated, once contextualized, would be rather small. Nevertheless also in this case we decided to keep our estimate on the upper side (also for consistency with the above mentioned perspective we want to introduce in the literature). The assumption here is that we do not know where most peatlands are. The soil type information the referee cites are also based on imprecise sources, as all sources in this particular context. In some cases they are based on surveys, in some other cases they are based on models, in others they are based on geostatistical techniques (e.g. the Soilgrid project). Considering such information as "true" represents a huge bias that is often not considered critically. Using the soil maps derived from the literature is one possible choice, but there might be a large error associated with such choice (we refer here also to our response to the first reviews where we provided a detailed comparison of different sources on area and allocation of organic soils).

Since we wanted to represent an alternative perspective, we included all the Ramsar wetland sites since we assumed that most of them could represent associated peatlands (due to the ecology and climate of most sites). Temperate peatlands would otherwise be virtually completely neglected in estimates based on other approaches (which are the ones in the literature the referee cites), but these peatlands could be instead quite important (see for example Ott et al., 2016). Even if its impact is quite small/negligible, still the inclusion of the Ramsar dataset is therefore in line with the choice of this manuscript of being willing to offer an upper

limit for the global peatland area estimate.

Ott, C. A., & Chimner, R. A. (2016). Long-term peat accumulation in temperate forested peatlands (*Thuja occidentalis* swamps) in the Great Lakes region of North America. *Mires and Peat*, 18(1), 1–9. <http://doi.org/10.19189/MaP.2015.OMB.182>

In any case, the present ms is lacking adequate documentation and an explanation for the much larger land base of peatland soils that is derived.

As we explained above the vast majority of the large base of peatland soils is derived from Yu et al., 2010, and this was already described in the manuscript. We nevertheless clarified it further.

Secondly, the overlay and then classification of peatland into various land use categories is not well done, but I believe could be improved. For example, in the Boreal climate zone, the largest land use category (over 25% of the global total!) is the ‘mosaic’ of combined cropland, grassland, forest. Even if much of the ESA-GCLM product in this area showed as code 30, it is clear that the area of what would be IPCC classified cropland is vanishingly small (maybe a bit of potatoes in northern Norway, some cabbages along the Peace River in Alberta ?!). That area has to be almost entirely a mixture of forest and ‘grassland’ bogs – I think you could refine and improve this classification using something like Ramankutty’s cropland land covers or even some regional agricultural statistics, to as much as possible differentiate between cropland and non-cropland peat areas.

Actually the climate classification had formerly been aggregated by country (since that is the aggregation level of the starting data about peatland degradation by Joosten 2010) based on the mode of the whole area of the country. In such classification there were just two countries with the “boreal” classification, Canada and Russia. Norway for example was, based on the mode, classified as temperate, as well as Sweden (and this is actually quite exact in the opinion of one of the authors, actually living and working in Sweden). According to the reviewers suggestion we now changed from a per-country scale to a pixel-based approach and use Köppens classification, which greatly improves the spatial resolution of climate zones needed for the assignment of emission factors. Further, the suggested map by Ramankutty (http://www.earthstat.org/wp-content/uploads/2014/08/PRINT50_CropPasture.png) is indeed a valuable product, and we are grateful to the referee for having brought it to our attention. We set up on such map a threshold of 50% probability of agricultural land, and utilized the derived map to integrate our formed land use classification. We believe that this greatly improved the robustness of our estimates. Thanks again for the suggestion! The improvement of spatial representation also made it necessary to change the type of projection of the land’s surface.

As a side remark to the comment, we stress that Northern countries are not the steppe that is often pictured in documentaries, but are rather places where many

millions of people live, cultivate and eat the food they produce. The whole area of Russia and Canada is indeed pretty rich in agricultural areas, see for example the state of Alberta:

(http://agriculture.alberta.ca/acis/maps/agricultural_land_resource_atlas_of_alberta/soil/soil_quality/cultivation_intenstity_index_%202001_big_map.png), which also has a rather good university of agriculture (<https://www.ualberta.ca/agriculture-life-environment-sciences/>) and presents much richer agriculture than just cabbage.

Along the same lines, the way peat degradation is assigned to land covers is poorly conceived and biases the results. Line 195 states ‘... based on the assumption that peatland degradation areas are randomly distributed among all the land use classes’. This impacts the results in Fig. 1 and Table 1. I would submit that you should assume that 100% of the cropland land area that is on peat soils, falls into the degraded peatland category – by definition! Growing annual crops on organic soils is the dominant degradation cause (excepting peat mining for energy) – i.e., you have to drain, lime, cultivate, etc (cranberry bogs aside). Even rice on organic soils – likely the only cropland use that doesn’t involve complete drainage – would lead to net losses of the organic matter. So the logical approach would be to first assign degraded organic soil area to cropland, and then perhaps assume an equal distribution of remaining degraded peatland area allocated between the grassland and forestland categories, unless other information indicates what systems are most likely to have had peatland drainage.

We are grateful to the referee for this really useful and precious suggestion. We admit we did not see this possibility before, and we really appreciate the possibility of improving our calculations and did so by including the cropland area as described above. We re-run our calculations based on the referee’s suggestion. Whenever the new land-use map indicated a cropland, we now set the degradation to 1 (corresponding to the whole area in that specific position being degraded). In case the agricultural area of a country covered a smaller proportion of the degraded peatland in such country, we assigned 1 to the peatlands on agricultural areas in such country and we redistributed the remaining proportion of degraded peatland per country uniformly among the other classes.

The new peat degradation map presents an updated estimate of the allocation of total degrading peatland, and (although such map is of course still uncertain in some parts) we believe that this improved the robustness of our calculations. We are grateful to the referee for the precious suggestion!

Accordingly there seem to be lots of problems with Table 1 – e.g., as noted above, for CL in Boreal and Tropical areas, most of the peat area under CL is shown as NOT degrading!;

This was due to the approach that we previously followed, that we now modified according to the referee’s suggestions. Thanks again.

Also in the CL temperate area there is no (0.00) land area shown as only ‘CL’ yet there is a degrading peat C stock assigned to that category along with calculated

GHG emissions; similarly, the tropical mosaic (CL+GL+FL) has no (0.00) area, but also a large degrading peat stock and GHG emissions.

Indeed there were problems! A wrong cell pointer in the excel file (pointing to an empty cell) produced that weird problem of no area as CL. The calculations are now completely re-done (also because of wanting to implement the referee's suggestions) and the problem solved. We apologize for the issue, and we are grateful to the referee to have pointed it out.

The way in which the degrading peatland area is assigned to land uses, will affect the relative weighting of the emission factors used (which differ by land use and climate) and hence the overall estimates of the GHG emissions. At present it difficult to have confidence in those estimates – and to evaluate the other secondary results that depend on the land cover/peatland allocations, given the outstanding methodological issues.

Again, we are grateful to the referee for the precious suggestions that improved our confidence in our map (and, we believe, its robustness). These kind of estimates present still a huge difficulty due to the associated uncertainty of every source involved at this scale, but we are convinced that the referee's suggestion improved our estimates for what is possible.

Reviewer #3 (Remarks to the Author):

Dear authors and editor,

I'm pleased by the changes in the manuscript regarding my main criticism about the peatland dataset. I'm impressed by the extensive methodological explanations and considerations provided from the authors in their response to the first manuscript review.

The methodology is now well explained; the peatland dataset is improved and especially figure 1 has been successfully enhanced.

The authors now provide reasonable and substantiated global estimates on peatland emissions, both, the current and the possible future emissions.

I recommend this manuscript for publishing in Nature Communications, because it is enhanced to appropriate methodological quality and appreciates the very important (and at global level underdeveloped) area of peatland research in context of climate change.

You may give my full name in the online documentation of the review process.

Thanks for the positive evaluation.

Review of revised Leifeld & Menichetti

The manuscript has been considerably improved from the previous version with the correction of several major errors in the data compilations and analyses, as described in the author responses to the previous review.

There's no question in my mind that the topic is important and will engender considerable interest in the scientific community as the issue of land use, GHG emissions, mitigation potential, etc. is one of the truly 'hot topics' at the moment. That is why I believe it is critical that the authors have a firm basis for their results and conclusion – you need to get it 'right'!

The major concern that I still have is with the derivation of the total areas for peatland which then drive the estimates of 'degraded' managed peatland and hence emission estimates and mitigation potential. You (the authors) state your intent to present an “*upper estimate of the possible global peatland area*” (lines 172-185). While I understand the sentiment (we think probably current peatland area estimates – e.g. the 440 Mha in Yu et al. Table 1; roughly similar to many other estimates since 1980s (see table below) – is an underestimate, but we don't know by how much), I think it is potentially very dangerous to then produce your new 'upper level' estimate, if it's not defensible! Certainly when your new estimate is 955 Mha – roughly double the area. It's a bold assertion and thus needs strong support.

Author	Year	Area (million ha)
Bulow	1929	110
Nikonow and Sluka	1964	112
Tibbets	1969	150
Moore and Bellamy	1974	230
Kivinen and Pakarinen	1980	420
BORD NA MONA	1984	420
Heikurainen	1982	500
Nasa	1987	530 (wetlands)
Mathews and Fung	1987	526 (wetlands)
Maltby	1988	500 (wetlands)
Aselman and Crutzen	1989	557 (wetlands)
Lappalainen	1994	398.5

Source: E. Lappalainen 1996, Bord Na Mona 1988, and M.N. Nikonov et al. 1964.

Table 1. Different authors' estimation of global peatland areas

©Encyclopedia of Life Support Systems (EOLSS)

From the description of your new estimates for total peatland (lines 170-185) it gives the impression that this increase is accounted for by the country-level updates and the Ramsar convention sites – however, at the end of that section you state that 'our updated peatland extension map accounted for an increase of (only) 0.8% compare to ⁴[the **Yu et al. map values**]'. Thus I conclude that the main basis

for your estimate of 955 Mha is the summation of the polygons in the peatland map of Yu et al. You quoted Yu et al. in your rebuttal, but you left out the most important statement that they made! (my emphasis added), from Yu et al., bottom of paragraph [4]: “*The peatland areas we used in the peat C pool calculations were derived from the literature (see Table 1 and the auxiliary material) rather than from the new peatland map presented. This is necessary because the peatland map shows peatland-abundant regions where peatlands cover at least 5% of the landmass, but accurate peatland coverage and distribution is not available for many mapped regions”.* My interpretation of this is that Yu et al. mapped a lot of area that was *only fractionally (>5%)* peat soils – hence their map shows *areal distribution* well but the mapped areas should **NOT** be added up and represented as the area of peatland. I think this is THE major concern in the present version of your manuscript, where you have this much larger area of peat soils than has ever been previously reported because of a flawed interpretation of the Yu et al. work. Thus the justification for that ca. doubling in the global peat soil area is well justified.

You also mention the recent Dargie et al. paper on peatlands in the tropics, as an example of previously unaccounted for peatland, which I’d agree with. However, Dargie et al. report a peatland area of 14.5 Mha and state that is ‘3X more’ the previous reported area, e.g. 2-3 Mha in past surveys. So that is an example of an addition 10 Mha on previously unmapped peatland – **but only a small fraction of the circa 500 Mha of additional peatland** that your paper is claiming relative to Yu et al. and many other previous estimates.

What I recommend is that you adjust your total peatland area starting from a suitable baseline – e.g., Yu et al. table values or similar estimates, and then add *additional* previously unaccounted peatland area from your country updates, the Dargie et al. study, the Ramsar sites, etc. – where you can document the increase in peat area that you can actually account for, in a new Table. It’s critical that your assumption and data sources be as transparent as possible. You could still use the Yu et al. map to do the land cover overlays to get **relative** values for each land cover and climate class, which could then be adjusted according to a revised tabular peatland area estimate.

Because total peatland area drives all the other calculations (degraded area, emissions, etc.) it has to have a solid foundation.

Other specific comments:

Line 78 and 81 - Line 78 states that the ‘lower range of this latter estimate’ is based on Yu et al. and then line 81 states ‘The upper range includes all the histosols data reported by Yu et al.’ Is that what you mean, both the lower and upper limits from the same source??

Line 83 – States that ‘...0.5-1.7 Gt would be released upon degradation of currently utilized tropical peat lands’. If they are currently utilized (i.e., drained, cultivated) now, wouldn’t they be emitting now (not ‘*would be*’ – i.e., at some time in the future)?

Line 88 – ‘our analysis suggests the magnitude of that sink is probably overvalued’. – Don’t you mean **undervalued**? (If there are larger than assumed terrestrial fluxes TO the atmosphere, then the net CO₂ flux to the land surface (sink) must be larger to achieve balance).

Table 1. I still find your figure of 13 Mha of annual cropland on peat soils in the boreal zone very large! Where is all that cropland? I made a similar comment in the previous review and referenced some minor agriculture perhaps in northern Norway and Alberta (as figurative examples) and your response was that you were classifying those areas as 'temperate', which is fine by me. But then if not in Scandinavia or Canada, is it all in N. Russia?? Are you confident of that estimate? It is a lot of area to be just in cropland of any type in the boreal zone, let alone all on peat soils! (By way of comparison, **total** annual cropland of Sweden and Finland are ca. 1 Mha each and almost all of that is in the southern parts of those countries and thus very much so in 'temperate' (and not boreal)).

Supl Fig. 1. You show a high ratio of degradation for peat soils in central Alaska. That is pretty much exclusively forested peatlands that are almost entirely unmanaged, in contrast to say forested peatlands in Finland and Sweden which your map correctly shows as 'degraded' because there is extensive drainage there on forest lands that managed for wood production – but that is NOT the case in Alaska! There are areas there subjected to wildfires and thus emissions from burning peatsoils but I don't believe you included that in the definition of degradation.

Review of revised Leifeld & Menichetti

The manuscript has been considerably improved from the previous version with the correction of several major errors in the data compilations and analyses, as described in the author responses to the previous review.

There's no question in my mind that the topic is important and will engender considerable interest in the scientific community as the issue of land use, GHG emissions, mitigation potential, etc. is one of the truly 'hot topics' at the moment. That is why I believe it is critical that the authors have a firm basis for their results and conclusion – you need to get it 'right'!

The major concern that I still have is with the derivation of the total areas for peatland which then drive the estimates of 'degraded' managed peatland and hence emission estimates and mitigation potential.

You (the authors) state your intent to present an "upper estimate of the possible global peatland area" (lines 172-185). While I understand the sentiment (we think probably current peatland area estimates – e.g. the 440 Mha in Yu et al. Table 1; roughly similar to many other estimates since 1980s (see table below) – is an underestimate, but we don't know by how much), I think it is potentially very dangerous to then produce your new 'upper level' estimate, if it's not defensible! Certainly when your new estimate is 955 Mha – roughly double the area. It's a bold assertion and thus needs strong support.

From the description of your new estimates for total peatland (lines 170-185) it gives the impression that this increase is accounted for by the country-level updates and the Ramsar convention sites – however, at the end of that section you state that 'our updated peatland extension map accounted for an increase of (only) 0.8% compare to 4[the Yu et al. map values]'. Thus I conclude that the main basis for your estimate of 955 Mha is the summation of the polygons in the peatland map of Yu et al. You quoted Yu et al. in your rebuttal, but you left out the most important statement that they made! (my emphasis added), from Yu et al., bottom of paragraph [4]: "The peatland areas we used in the peat C pool calculations were derived from the literature (see Table 1 and the auxiliary material) rather than from the new peatland map presented. This is necessary because the peatland map shows peatland abundant regions where peatlands cover at least 5% of the landmass, but accurate peatland coverage and distribution is not available for many mapped regions". My interpretation of this is that Yu et al. mapped a lot of area that was only fractionally (>5%) peat soils – hence their map shows areal distribution well but the mapped areas should NOT be added up and represented as the area of peatland. I think this is THE major concern in the present version of your manuscript, where you have this much larger area of peat soils than has ever been previously reported because of a flawed interpretation of the Yu et al. work. Thus the justification for that ca. doubling in the global peat soil area is well justified.

You also mention the recent Dargie et al. paper on peatlands in the tropics, as an example of previously unaccounted for peatland, which I'd agree with. However, Dargie et al. report a peatland area of 14.5 Mha and state that is '3X more' the previous reported area, e.g. 2-3 Mha in past surveys. So that is an example of an addition 10 Mha on previously unmapped peatland – but only a small fraction of the circa 500 Mha of additional peatland that your paper is claiming relative to Yu et al. and many other previous estimates.

What I recommend is that you adjust your total peatland area starting from a suitable baseline – e.g., Yu et al. table values or similar estimates, and then add additional previously unaccounted peatland area from your country updates, the Dargie et al. study, the Ramsar sites, etc. – where you can document the increase in peat area that you can actually account for, in a new Table. It's critical that your assumption and data sources be as transparent as possible. You could still use the Yu et al. map to do the land cover overlays to get relative values for each land cover and climate class, which could then be adjusted according to a revised tabular peatland area estimate.

Because total peatland area drives all the other calculations (degraded area, emissions, etc.) it has to have a solid foundation.

First, we want to point out that we definitely concur with the referee that a proper area estimate is a rather crucial and difficult point and we really appreciate the time and energies spent for getting it as precise as possible.

We agree now with the referee's position, and followed his/her advice by applying a correction in sensu Yu et al. to our estimate in order to rescale the mapped values to a more conservative area estimate. The new area estimate used for the calculation of stocks and area distribution (464 Mha) comes from the aggregation of Yu et al. reported value (in their table and not their map) updated with values reported by Page et al. 2011 (cited in text) for tropical peatlands (from 36 Mha in Yu et al. to 44 Mha in Page et al 2011) and with values reported by Dargie et al. 2016 (cited in text) (area reported for Congo basin, around 16 Mha).

This area, together with South American basins, are likely to still leave out some uncharted important peatland deposits, but we agree that the error we commit like this is probably less than including in such calculations, whenever data are not available, histosols and gleysols as done in the map from Yu et al. In order to avoid the risk of misinterpretation of the uncertainty bounds and not to offer a misleading impression to the readers, we therefore followed the referee's suggestion.

Other specific comments:

Line 78 and 81 - Line 78 states that the 'lower range of this latter estimate' is based on Yu et al. and then line 81 states 'The upper range includes all the histosols data reported by Yu et al.' Is that what you mean, both the lower and upper limits from the same source??

Mostly yes, since the area reported in the tables from Yu et al. is much smaller than the area included in their map. That source includes therefore, in our view, already quite a relevant uncertainty. Still the upper estimate included also the area we added compared to Yu et al., which is anyway quite small in proportion.

Nevertheless the revised manuscript was changed to reflect the new approach the referee suggested and, hence, the comment is obsolete now.

Line 83 – States that '...0.5-1.7 Gt would be released upon degradation of currently utilized tropical peatlands'. If they are currently utilized (i.e., drained, cultivated) now, wouldn't they be emitting now (not 'would be' – i.e., at some time in the future)?

The statement refers to the cumulative emissions to be expected from those peatlands until they are fully exhausted, not to the current, i.e. annual rate of emission. Reworded in text.

Line 88 – 'our analysis suggests the magnitude of that sink is probably overvalued'. – Don't you mean undervalued? (If there are larger than assumed terrestrial fluxes TO the atmosphere, then the net CO₂ flux to the land surface (sink) must be larger to achieve balance).

The reviewer is right and we changed the wording accordingly.

Table 1. I still find your figure of 13 Mha of annual cropland on peat soils in the boreal zone very large! Where is all that cropland? I made a similar comment in the previous review and referenced some minor agriculture perhaps in northern Norway and Alberta (as figurative examples) and your response was that you were classifying those areas as 'temperate', which is fine by me. But then if not in Scandinavia or Canada, is it all in N. Russia?? Are you confident of that estimate? It is a lot of area to be just in cropland of any type in the boreal zone, let alone all on peat soils! (By way of comparison, total annual cropland of Sweden and Finland are ca. 1 Mha each and almost all of that is in the southern parts of those countries and thus very much so in 'temperate' (and not boreal)).

Part of that number was due to the larger peatland area we formerly chose to base our estimate upon, and now that value is halved (as pretty much all the others). But we are rather confident about

this latter estimate. Regarding the assignment to climate zones it is important to notice that the climatic areas we base our calculation on are ecological, not geographical (it is a revised Köppen classification). This classification is more detailed than, for example, the one proposed by IPCC. The areas reported in the appendix, in our climate and land use map, reveal how based on this classification most of Sweden, Norway and Finland, as well as most of Canada and Russia, are classified under these ecological criteria as “boreal” (all assigned to “D” climates). Actually the whole Finland is classified as boreal (either Dfc or Dfb classes).

Former estimates (revision 1) were based on the same map (although a less updated version), but we assigned the climate classification by country based on a mode estimate and this was producing the distortions we mentioned in our previous revision. Current estimates are per pixel, and only the ecological distribution of climates now counts. This is a great improvement and we cannot think of any way to make it better.

In order to have an idea of how the climatic zones are assigned by Peel et al., 2007 (cited in text), we refer to Supplementary Figure 2.

Supl Fig. 1. You show a high ratio of degradation for peat soils in central Alaska. That is pretty much exclusively forested peatlands that are almost entirely unmanaged, in contrast to say forested peatlands in Finland and Sweden which your map correctly shows as ‘degraded’ because there is extensive drainage there on forest lands that managed for wood production – but that is NOT the case in Alaska! There are areas there subjected to wildfires and thus emissions from burning peatsoils but I don’t believe you included that in the definition of degradation.

Many thanks for pointing this mistake out. The estimate for Alaska come from having aggregated the data on a country basis. This process assigned to Alaska the same degradation ratio than U.S. This conceptual mistake is now amended, and Alaska is considered separately. Calculations have been updated accordingly. Again thank you for offering us this further occasion for improvement.

Review of revised MS by Leifeld and Menichetti

I appreciate the comments to my previous review regarding the methodology used to estimate peatland areas and calculations made to estimate the distribution of peatland area by land use classes which then go into the estimates of current C stock loss from degrading peatlands and potential future losses from further peatland exploitation. The total peatland area shown in the revised Table 1 (463 Mha) now seems to be more in line with previous published estimates – in any case the authors seem to agree that there was not sufficient evidence to support the much higher estimate of 955 Mha that was given in the previous version. That was the main concern that I had in the previous review.

However, now I am very puzzled that this large decrease in the estimated area of total peatland, which now shows much reduced areas for each of the land use classes within each of the climate zones, seems to have minimal effect (in some cases nil effect) on the emission estimates that are the core of the paper. My understanding from the methodology is that the emissions are derived from a stratified (landuseXclimate) set of emission factors multiplied by the corresponding stratified land areas. Thus I cannot understand that the emission estimates should be **insensitive** to these large changes in peatland areas and thus peatland C stocks!

For example, the abstract has the same estimate for cumulative C and N emissions in the 3rd and 4th (current) MS versions: 80.8 GtC, 2.3 GtN (although the new version no longer gives a range (e.g. 33.1-80.8 GtC) – I'm not sure why that change and I don't think is an improvement.

Ln 26 – the cited total C stock of organic soils changed very little with the revision (667 GtC v3; 644 GtC v4), despite the halving of the area.

In Table 1 of the previous version (3), the total area of degrading peatlands, for each climate zone, as well as the estimated C emissions from those degrading lands, are **exactly the same** in the new version of Table 1, despite there being ½ the total area of peatland in the previous version. Given that the methodology for determining the area of degrading peatland is exactly the same in both versions (section 2.3), I don't see how it is possible the total area of degraded peatland could remain unchanged while the area estimates for total peatland in each climate and land use categories decreased substantially with the revision (see below).

Table from version 3 of the manuscript

441 *Table 1. Land use, area, annual GHG emissions from degrading peatlands worldwide, and their carbon pool size. Emissions include CO₂, CH₄, N₂O and DOC.*

443

Climate	Total peatland	CL ^a	GL ^a	FL ^a	GL/FL ^a	CL/GL/FL	Degrading peatland	Actual Emissions ^b	Peat C	Degrading peat C
----- Area (Mha) -----								Gt CO ₂ eq.	----- Gt C -----	
Tropical	186.6	27.0	35.9	110.0	5.0	8.6	24.2	1.49 (0.04-2.79)	104.7 ^d – 379.1 ^e	13.6 - 49.1
Temperate	35.2	6.6	9.5	16.7	1.3	1.1	10.6	0.16 (0.1-0.21)	25.7 ^d – 41.6 ^e	7.7 - 12.5
Boreal	685.8	12.9	162.7	474.1	34.1	2.1	15.5	0.28 (0.17-0.4)	501.5 ^d – 811.3 ^e	11.3 - 18.3
Polar	47.5	0.3	28.2	18.3	0.1	0.6	0.7	0.01 (0-0.02)	34.8 ^d – 56.2 ^e	0.5 - 0.8
Oceanic	0.1	0.0	0.0	0.0	0.0	0.0	0	0 (0-0)	NA	NA
Total	955.2						50.9	1.93 (0.31-3.42)	666.7 – 1288.3	33.1 - 80.8

444 ^a Peatland area distributed by land use, land-use classes are cropland (CL), grassland (GL) and forest
 445 land (FL). Rows with more than one land-use represent areas where a clear assignment to one type is
 446 not possible (see methods). ^b Annual means, values in parentheses express the range from minimum
 447 to maximum. ^c based on ⁶. ^d based on ⁸. ^e calculated by applying C densities in ⁴ and ⁸ to our newly
 448 delineated areas.

Table from version 4 (current) of the manuscript

467 *Table 1. Land use, area, annual GHG emissions from degrading peatlands worldwide, and their carbon pool size. Emissions include CO₂, CH₄, N₂O and DOC.*

469

Climate	Total peatland	CL ^a	GL ^a	FL ^a	GL/FL	CL/GL/FL	Degrading peatland	Actual Emissions ^b	Peat C	Degrading peat C
----- Area (Mha) -----								Gt CO ₂ eq.	----- Gt C -----	
Tropical	58.7	8.5	11.3	34.6	1.6	2.7	24.2	1.48 (0.04-2.79)	119.2 ^{c,e}	49.1
Temperate	18.5	3.5	5.0	8.9	0.7	0.6	10.6	0.16 (0.10-0.21)	21.9 ^{d,e}	12.5
Boreal	360.9	6.8	85.6	249.5	17.9	1.1	15.5	0.26 (0.16-0.36)	427.0 ^{d,e}	18.3
Polar	25.0	0.1	14.9	9.7	0.1	0.3	0.7	0.01 (0-0.02)	29.6 ^{d,e}	0.8
Oceanic	<0.1	<0.1	<0.1	<0.1	<0.1	<0.1	<0.1	0 (0-0)	<0.1	<0.1
Total	463.2						50.9	1.91 (0.31-3.38)	597.8	80.8

470 ^a Peatland area distributed by land use. Land-use classes are cropland (CL), grassland (GL) and forest
 471 land (FL). Rows with more than one land-use represent areas where a clear assignment to one type is
 472 not possible (see methods). ^b Annual means, values in parentheses express the range from minimum
 473 to maximum. ^c based on ^{5,6}. ^d based on ⁸. ^e assignment to climate region by applying C densities in ^{5,6,8}
 474 to our newly delineated areas.

Author response to reviewer comment NCOMMS-16-14037C

We thank the reviewer for studying our manuscript once more. In the following we provide a point-by-point response to his/hers comments.

Reviewer: The total peatland area shown in the revised Table 1 (463 Mha) now seems to be more in line with previous published estimates – in any case the authors seem to agree that there was not sufficient evidence to support the much higher estimate of 955 Mha that was given in the previous version. That was the main concern that I had in the previous review.

Authors: Yes, we agree that the previous max. area of 955 Mha seems difficult to defend and we followed the suggestion of the reviewer.

Reviewer: However, now I am very puzzled that this large decrease in the estimated area of total peatland, which now shows much reduced areas for each of the land use classes within each of the climate zones, seems to have minimal effect (in some cases nil effect) on the emission estimates that are the core of the paper. My understanding from the methodology is that the emissions are derived from a stratified (landuseXclimate) set of emission factors multiplied by the corresponding stratified land areas. Thus I cannot understand that the emission estimates should be insensitive to these large changes in peatland areas and thus peatland C stocks!

Authors: As described in the methods section, the estimate of the degradation factor for peatlands is mostly based on Joosten's (2010) estimate of the fraction of degrading peatlands per country. That estimate did not change from revision 3 to 4 (see Table). We consider it accurate since it very well reflects the state in those countries where degradation is particularly severe and peat extension, at the same time, is most certain. These areas are also already considered in our initial peatland map. After we multiply the degradation raster with the potential emission raster, those peatland areas contributing by far the most to the overall emissions (i.e., the degrading ones) end up being congruent with those where we can trust our extension map the most (and this is the reason why our degradation estimate does not differ that much from Joosten, 2010 although he did not provide an assignment to climate or land-use type). To put in other words –if, for example, we set the whole area of Peru, where there is almost no degradation going on, to peat, this would not change its degrading area. Therefore neither the emissions nor the area of degrading peatlands needed to be rescaled between version 3 and 4, but we corrected the total peatland area (see response to comment above).

The increase relative to the first submission was mostly due to an improvement in the cartographic projection of the earth. Further, we use here a refined methodology, also in response to previous comments from other reviewers, by including the global cropland estimate from Ramankutty et al. (2008), considering all organic soils cropped for agriculture to be drained. Below we summarize our estimates on degraded area and overall GHG emissions in the iterative versions of our manuscript. It can be seen, that degraded peatland area remained relatively constant.

Submission date	degraded area (Mha)	GHG emission (Gt CO ₂ -eq.)	iteration
15 June 2016	45.7	1.22 (0.63-2.16)	original
4 Nov. 2016	46.1	1.57 (0.27 – 2.90)	new area data included, improved land-use assignment
7 April 2017	50.9	1.93 (0.31 – 3.42)	cropland explicitly allocated, new climate classification, refined cartographic projection
8 August 2017	50.9	1.91 (0.31 – 3.38)	revision of total peatland extension, improved stock estimate (Loisel, Page, Dargie)

Reviewer: For example, the abstract has the same estimate for cumulative C and N emissions in the 3rd and 4th (current) MS versions: 80.8 GtC, 2.3 GtN (although the new version no longer gives a range (e.g. 33.1- 80.8 GtC) – I’m not sure why that change and I don’t think is an improvement

Authors: The cumulative C and N emissions do not change as, indeed, the degraded area remains the same between version 3 and 4 and was slightly moved upwards between revisions 1, 2 and the more recent ones (see Table). We provided a range for the amount of C in degraded peatlands in version 3. Its upper limit (80.8) was based on the current (version 4) carbon stock of 1587 t C/ha (average of data in Loisel, Dargie, Page weighed by contribution of temperate/boreal vs. tropical) times the degraded area. The lower estimate (33.1 Gt C) was based on the assumption that the upper area estimate for total peatland in that version (c. 955 Mha) comes along with the lower estimate for global carbon stock in all peatland based on Yu et al. (2010, resulting in a per-hectare stock of only c. 700 t. This is, from the viewpoint of the publications we now decided to be the most reliable ones when it comes to per-area carbon stocks, namely Loisel, Dargie, Page (all cited in the MS), and from the viewpoint of our area-revision between version 3 and 4, a strong underestimation of per-area carbon stocks. We therefore consider this former lower estimate obsolete.

Reviewer: Ln 26 – the cited total C stock of organic soils changed very little with the revision (667 GtC v3; 644 GtC v4), despite the halving of the area.

Authors: In our study, the global stock estimates derive from literature data (see Table 1 in MS and methods). Our third version from April 2017 assigned an overall peatland area of c. 955 Mha. This area was, however, not used for calculating total stocks but for displaying the area extent of peatlands in our maps. That extent and allocation was/is needed to assign climate classes to the respective areas and, hence, to adjust emissions.

Reviewer: In Table 1 of the previous version (3), the total area of degrading peatlands, for each climate zone, as well as the estimated C emissions from those degrading lands, are exactly the same in the new version of Table 1, despite there being ½ the total area of peatland in the previous version. Given that the methodology for determining the area of degrading peatland is exactly the same in both versions (section 2.3), I don’t see how it is possible the total area of degraded peatland could remain unchanged while the area estimates for total peatland in each climate and land use categories decreased substantially with the revision (see below).

Authors: As also described above and in section 2.3 of version 4 our manuscript, the area of degraded peatlands is dependent mostly on Joosten (2010) estimates. We write: “We then calculated a degradation ratio (between 0, non-degraded, to 1, fully degraded) using the area of degraded peatland by country reported by Joosten (2010).” We further elaborate that in case the area of cropland on or-

ganic soil, for the respective country, should exceed the absolute area of degraded peatland given by Joosten (2010), the overall degraded area per country would increase correspondingly. That's why our degraded area of 50.9 Mha slightly exceeds the estimate of Joosten (2010) (46.4 Mha). We consider this an improvement.

Dear editor and authors,

I have reviewed the manuscript before and have now been asked “to assist by commenting on the authors' response to Reviewer 1's comments, and whether I find the reviewer's comments fair and the authors' response appropriate.” I will not comment on ‘fairness’ because I think it is a misunderstanding.

As far as I can see, Reviewer 1 has not comprehended the methodology completely - especially which dataset has been used for which purpose. **His main concern behind almost all raised points is the reduction of the overall peatland area from c. 955 Mha to 463 Mha between different versions of the manuscript and the implications he assumed.** He asked:

- Why the area of degrading peatland has not reduced accordingly?
- Why the abstract has the same estimate for cumulative C and N emissions as before and why there is no range anymore?
- Why the cited total C stock of organic soils changed very little with the revision despite the halving of the area?
- How it is possible the total area of degraded peatland could remain unchanged while the area estimates for total peatland in each climate and land use categories decreased substantially with the revision?

The answer to these questions is that the degrading peatland areas have been deduced from Joosten et al. (2010) in all manuscript versions (with minor changes), and the peatland extension map has been utilized only as a statistically representative geographical sample to recalculate proportions of emissions and land use types. **In other words: There is no direct relationship between the area of degrading peatland (cf. Joosten et al 2010) and the overall peatland area (compiled in this study) and thus, the considerable decrease of the overall peatland area has no influence on the degrading peatland area.**

Generally, I consider the authors response appropriate.

Kind regards,
Alexandra Barthelmes!

Reviewer:

Dear editor and authors,

I have reviewed the manuscript before and have now been asked “to assist by commenting on the authors' response to Reviewer 1's comments, and whether I find the reviewer's comments fair and the authors' response appropriate.” I will not comment on ‘fairness’ because I think it is a misunderstanding. As far as I can see, Reviewer 1 has not comprehended the methodology completely - especially which dataset has been used for which purpose. His main concern behind almost all raised points is the reduction of the overall peatland area from c. 955 Mha to 463 Mha between different versions of the manuscript and the implications he assumed. He asked:

- Why the area of degrading peatland has not reduced accordingly?
- Why the abstract has the same estimate for cumulative C and N emissions as before and why there is no range anymore?
- Why the cited total C stock of organic soils changed very little with the revision despite the
- halving of the area?
- How it is possible the total area of degraded peatland could remain unchanged while the area estimates for total peatland in each climate and land use categories decreased substantially with the revision?

The answer to these questions is that the degrading peatland areas have been deduced from Joosten et al. (2010) in all manuscript versions (with minor changes), and the peatland extension map has been utilized only as a statistically representative geographical sample to recalculate proportions of emissions and land use types. In other words: There is no direct relationship between the area of degrading peatland (cf. Joosten et al 2010) and the overall peatland area (compiled in this study) and thus, the considerable decrease of the overall peatland area has no influence on the degrading peatland area. Generally, I consider the authors response appropriate.

Kind regards,

Alexandra Barthelmes!

Authors: We are grateful for the constructive feedback. The referee did not ask for further revision of our manuscript.